# SMARCAD1 ATPase activity is required to silence endogenous retroviruses in embryonic stem cells

Parysatis Sachs[1], Dong Ding[1], Philipp Bergmaier[1], Boris Lamp[2], Christina Schlagheck[1], Florian Finkernagel [3], Andrea Nist[2], Thorsten Stiewe [2] & Jacqueline E. Mermoud[1]

Endogenous retroviruses (ERVs) can confer benefits to their host but present a threat to genome integrity if not regulated correctly. Here we identify the SWI/SNF-like remodeler SMARCAD1 as a key factor in the control of ERVs in embryonic stem cells. SMARCAD1 is enriched at ERV subfamilies class I and II, particularly at active intracisternal A-type particles (IAPs), where it preserves repressive histone methylation marks. Depletion of SMARCAD1 results in de-repression of IAPs and adjacent genes. Recruitment of SMARCAD1 to ERVs is dependent on KAP1, a central component of the silencing machinery. SMARCAD1 and KAP1 occupancy at ERVs is co-dependent and requires the ATPase function of SMARCAD1. Our findings uncover a role for the enzymatic activity of SMARCAD1 in cooperating with KAP1 to silence ERVs. This reveals ATP-dependent chromatin remodeling as an integral step in retrotransposon regulation in stem cells and advances our understanding of the mechanisms driving heterochromatin establishment.

---

[1] Institute of Molecular Biology and Tumour Research, Philipps University Marburg, Marburg 35043, Germany. [2] Institute of Molecular Oncology, Genomics Core Facility, Philipps University Marburg, Marburg 35043, Germany. [3] Institute of Molecular Biology and Tumour Research, Center for Tumor Biology and Immunology, Philipps University Marburg, Marburg 35043, Germany. These authors contributed equally: Parysatis Sachs, Dong Ding. Correspondence and requests for materials should be addressed to J.E.M. (email: mermoud@imt.uni-marburg.de)

Transposable elements (TEs), originally described as controlling elements by Barbara McClintock in 1950s, are now understood as functional components of genomes. One of the most exciting characteristics of TEs is their potential to regulate cellular gene expression. They play important roles in early mammalian development, including placentation and pluripotency. Moreover, they can rewire gene regulatory networks and impact on evolution[1–3].

TEs are distributed throughout mammalian genomes, comprising the largest fraction of their DNA. The majority are retrotransposons, which propagate through an RNA intermediate. These are either flanked by long-terminal direct repeats (LTR), as exemplified by endogenous retroviruses (ERVs), or lack LTRs, such as long and short interspersed nuclear elements (LINEs and SINEs). ERVs account for 8–10% of human and mouse genomes. Remnants of germ-line retroviral infections, they can be divided into three classes based on sequence similarity to exogenous retroviruses[4]. ERV class II intracisternal A-particles (IAPs) are among the most active mobile elements in the mouse, responsible for about 10% of all spontaneous mutations[5]. Most retrotransposons have accumulated mutations that render them incapable of transposition. Yet, their influence on the host genome is substantial, given their capability to serve as promoters, enhancers, or repressors[2,6]. Therefore, tight control of retrotransposon activity is essential to protect genome and transcriptome integrity. Indeed, disruption of ERV regulation has been linked to cancer and neurological disorders[7,8].

In embryonic stem cells (ESCs) retrotransposon activity is limited by the locus-specific establishment of a transcriptionally silent chromatin environment within a relatively open chromatin context[3,6,9]. One repressive histone modification that stands out is methylation of histone 3 at lysine 9 (H3K9), which is associated with a broad range of retrotransposons[10–15]. The KRAB associated protein 1, KAP1 (TRIM28; TIF1β), is a key component of the retrotransposon silencing machinery[6,9,12,16,17]. Docking of KAP1 at ERVs of classes I and II triggers the formation of H3K9me3 marked heterochromatin through the recruitment of the H3K9 histone methyltransferase SETDB1 (ESET) and co-repressor proteins like heterochromatin protein 1 (HP1)[11,12,15,17–19]. KAP1-SETDB1-mediated repression of ERVs preserves the transcriptional landscape of ESCs by preventing enhancer/promoter effects originating from these elements. Accordingly, depletion of KAP1 or SETDB1 in ESCs results in de-repression of multiple ERVs and genes in their vicinity[11–13,15,16,20–22].

ATP-dependent chromatin remodeling complexes use ATP hydrolysis to change chromatin structure and regulate accessibility[23]. The importance of these remodeling enzymes in the regulation of gene expression is widely accepted, but little is known about their contribution to the control of TEs. In pluripotent stem cells SNF2 helicase family members such as CHD5 (chromodomain helicase DNA binding protein 5) and ATRX (α-thalassaemia/mental retardation syndrome X-linked) have been implicated in the control of class III MERVL and class II IAP elements, respectively[9,24]. However, it remains unclear whether their remodeling activity plays a role in this context. Open questions also concern, which specific steps in the silencing process require prior or concurrent chromatin remodeling.

The SWI/SNF-like chromatin remodeler SMARCAD1 has emerged as an attractive candidate for controlling retrotransposon activity. Our proteomic analysis revealed KAP1 to be robustly associated with SMARCAD1 in mouse ESCs (mESCs)[25]. SMARCAD1 is characterized by a conserved SNF2-type ATPase/helicase domain and two CUE (coupling of ubiquitin to ER-degradation) domains. The first CUE domain (CUE1) mediates the direct interaction with KAP1[25]. Here we set out to determine whether the SMARCAD1-KAP1 interaction impacts retrotransposon silencing in mESCs.

Our genome-wide profiling of binding sites for SMARCAD1 reveals that it is predominantly associated with repressed chromatin states in ESCs, among them ERVs belonging to classes I and II. This study identifies SMARCAD1 as a component of the machinery that silences ERVs. We reveal that the catalytic activity of SMARCAD1 ensures the robust transcriptional repression of IAPs and nearby genes. Mechanistically, we show that one of the first steps in setting up heterochromatin at ERVs, namely the stable association of KAP1, requires a functional ATPase domain in SMARCAD1. Our results highlight a key role for SWI/SNF-like chromatin remodeling activities in the establishment of ERV silencing in mammals.

## Results

**SMARCAD1 is enriched in heterochromatin in mESCs.** SMARCAD1 is highly expressed in the inner cell mass of the mouse blastocyst[26,27]. Accordingly, we detect higher SMARCAD1 protein levels in mESCs than in embryonic or adult fibroblasts (Fig. 1a). To investigate potential roles for SMARCAD1 in ESCs, we generated stable cell lines in which normal levels of SMARCAD1 were reduced using small hairpin RNAs (Fig. 1b and Supplementary Figure 1a). SMARCAD1-deficient cells displayed impaired proliferation (Fig. 1c and Supplementary Figure 1b), but unaltered cell cycle profiles (Supplementary Figure 1c) and no overt increase in cell death (Supplementary Movies 1, 2). In addition, upon *Smarcad1* knockdown ESCs lost their typical morphology (Supplementary Figure 1d; quantification in Supplementary Figure 1e) and showed a reduction of the stem cell marker alkaline phosphatase (Supplementary Figure 1d)[27,28]. Collectively, these observations emphasize an important contribution of SMARCAD1 to mESC homeostasis. This prompted us to develop a robust SMARCAD1 chromatin immunoprecipitation protocol followed by DNA sequencing (ChIP-seq) to gain insights into where in the mESC genome SMARCAD1 functions.

We initially generated ESCs that express FLAG-tagged SMARCAD1 (Supplementary Figure 1f, g). Indirect immunofluorescence verified that the localization of tagged SMARCAD1 mimics the localization of the endogenous protein, namely a wide distribution throughout the nucleoplasm as well as association with DAPI-dense staining heterochromatin (Supplementary Figure 1g). FLAG ChIP-seq led to the identification of candidate SMARCAD1-binding sites in the ESC genome and allowed us to validate SMARCAD1 antibodies in ChIP-qPCR experiments (Supplementary Figure 2a, b). Moreover, a comparison of single- and double-cross-linking procedures showed the combined use of disuccinimidyl glutarate (DSG) and formaldehyde results in significantly improved enrichment of SMARCAD1 signals (Supplementary Figure 2c). We went on to perform ChIP-seq with an antibody against the endogenous protein on double-cross-linked wild-type (WT) ESCs, and, to correct for background signals, on cells where SMARCAD1 was depleted (KD) (Supplementary Table 1). We identified 5727 regions of specific SMARCAD1 enrichment absent from the SMARCAD1 knockdown ESCs. There was significant overlap between the FLAG and SMARCAD1 ChIP-seq datasets with 2380 overlapping peaks which we consider high-confidence binding sites (Intersection Fig. 1e; Supplementary Figure 2d, e). The majority of these sites reside in intergenic regions (Fig. 1d).

Next, we determined whether SMARCAD1 binding sites coincide with particular histone modifications. We performed ChIP-seq for H3K9me3 in the ESC line in which the SMARCAD1 ChIP was performed (Supplementary Table 1) and analysed our genome-wide datasets in conjunction with available methylation

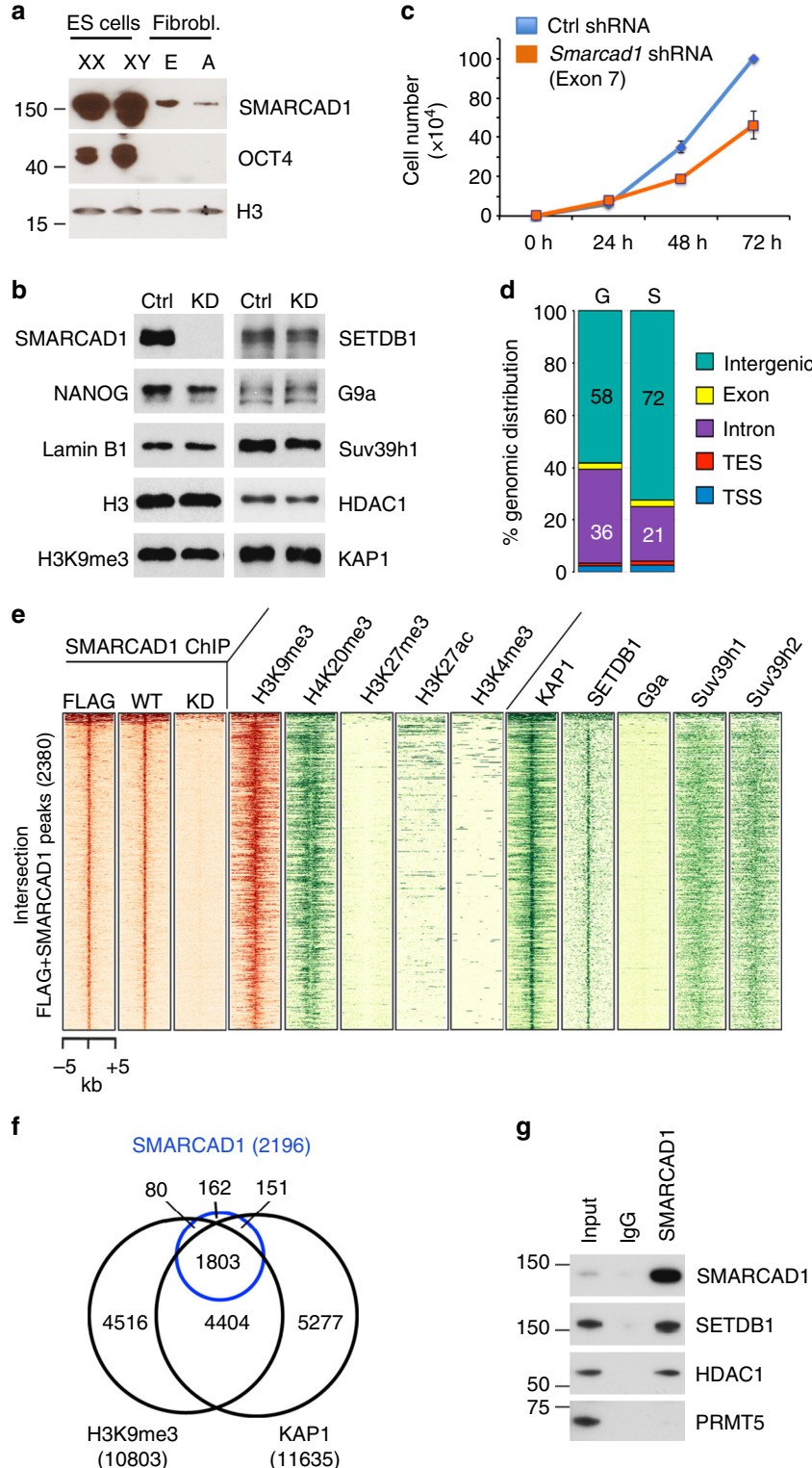

and acetylation profiles for H3 and H4 (Fig. 1e, middle panels). This comparison revealed a striking co-localization of SMAR-CAD1 with H3K9me3 and H4K20me3. Principally, these modifications mark pericentric heterochromatin and transposons, and are critical for keeping non-genic or lineage-inappropriate transcription in check[10,29]. While the vast majority of SMAR-CAD1 peaks (85%) overlaps with H3K9me3 (Supplementary Figure 2f), we observed minimal co-localization with the repressive mark H3K27me3, or with H3K27ac and H3K4me3,

marks of active enhancers or promoters (Fig. 1e, middle panels). These data show that SMARCAD1 is associated with repressed chromatin domains characterized by H3K9me3 and H4K20me3 in mESCs.

We went on to determine the fraction of SMARCAD1 enriched loci which are bound by KAP1 and found a very high degree of overlap (Fig. 1e, right). The majority of high confidence SMARCAD1 peaks (87%) coincide with KAP1 peaks (Supplementary Figure 2g). A three-way comparison of SMARCAD1,

**Fig. 1** SMARCAD1 associates with heterochromatin in mESCs. **a** Western blot showing SMARCAD1 levels in pluripotent mouse cells (XX PGK12.1 and XY E14 ESCs), embryonic (E) and adult (A) fibroblasts. OCT4 confirms the pluripotent state, histone H3 serves as a loading control. **b** Knockdown of SMARCAD1 in mESCs. Western blot was performed on control (Ctrl) and *Smarcad1* knockdown (KD) PGK12.1 cell extracts using antibodies specific for indicated proteins and modifications. Lamin B1 served as a loading control. MW markers are shown in Supplementary Figure 11. **c** Reduced proliferation of stable SMARCAD1 knockdown ESCs. Growth curves of PGK12.1 cells represent the mean ± SD from technical triplicates. Additional proliferation assays are shown in Supplementary Figure 1b. **d** Genomic distribution of SMARCAD1 binding sites (S) in comparison to the complete mouse genome (G). Data represent the intersection of the FLAG and SMARCAD1 ChIP (2380 regions) in PGK12.1 ESCs. TSS and TES correspond to transcriptional start site (−300 bp) and end sites (±500 bp). SMARCAD1 binds predominantly at intergenic sites (72%), with 2.4% exon, 21% intron, 1.6% TES, and 2.7% TSS. **e** Heatmap showing SMARCAD1 binding regions (±5 kb) corresponding to the intersection of the FLAG and SMARCAD1 ChIP-seq (2380 regions) in PGK12.1 ESCs. All regions were centered to the summit of the SMARCAD1-WT peaks and were sorted in descending order of signal intensity. Left, FLAG ChIP-seq and SMARCAD1 ChIP-seq data from WT and knockdown (KD) cells. Center and right panels show ChIP-seq intensity levels for histone marks and indicated chromatin proteins respectively, revealing SMARCAD1 co-localization with ERV-specific heterochromatic features. ChIP-seq profiles produced in this study are displayed in red, published datasets in green. **f** The vast majority (82%) of SMARCAD1 peaks are shared between KAP1 and H3K9me3. Venn diagram comparing the SMARCAD1 binding sites (Interesection of FLAG and SMARCAD1; 2196 peaks) with the H3K9me3 profile (10,803 peaks; this study) and KAP1 binding sites (11,635 peaks[34]). **g** SMARCAD1 is associated with SETDB1 and HDAC1 in ESCs. A SMARCAD1-specific antibody co-immunoprecipitates SETDB1 and HDAC1 from mESC nuclear extracts in the presence of benzonase and ethidium bromide, which prevents DNA mediated interactions. PRMT5 is a negative control. Lane 1, 3% input; lane 2, IgG; lane 3, IP

H3K9me3 and KAP1 ChIP-seq data reveals that most SMARCAD1 sites are shared with KAP1 and marked by H3K9me3 (Fig. 1f).

Methyltransferases with specificity for H3K9 include SETDB1, SUV39H1, SUV39H2 and G9a/GLP, all of which co-purify with KAP1[30]. Plotting the ChIP-seq signals of these enzymes with respect to SMARCAD1 binding sites revealed substantial overlap of SETDB1 with SMARCAD1 (Fig. 1e, right panels). Size exclusion chromatography of nuclear extracts showed co-elution of SMARCAD1, KAP1 and SETDB1 in high-molecular weight fractions (Supplementary Figure 2h). We confirmed the association of SMARCAD1 with SETDB1 by demonstrating that a SMARCAD1-specific antibody co-immunoprecipitated SETDB1, as well as the histone-deacetylase HDAC1 (Fig. 1g) and KAP1[25]. Altogether, our data point to a heterochromatin function for SMARCAD1 in mESCs.

**SMARCAD1 binds classes I and II ERV elements in mESCs.** The prominent co-localization of SMARCAD1 with H3K9me3 and known retrotransposon silencing factors exemplified by KAP1 and SETDB1 suggests that SMARCAD1 may participate in the regulation of TEs. Indeed, SMARCAD1 homologs in fission and budding yeast, like SMARCAD1 in mESCs, have previously been noted to accumulate at LTRs[27,31,32]. We therefore focused our analysis on retroelements and mapped our ChIP-seq data to the repeat database supplied by the UCSC Table Browser[33]. We included both unique and multi-aligned reads in our analysis. Reads which mapped equally well to multiple positions were aligned to only one of its best hits[34]. We found that SMARCAD1 occupies LTR retrotransposons belonging to ERV subfamilies of classes I and II (Fig. 2a). Two-independent ChIP-seq experiments, conducted with either a FLAG or a SMARCAD1 antibody, gave essentially identical results (Supplementary Figure 3a). The enrichment over these elements does not occur in SMARCAD1 knockdown cells (Supplementary Figure 3a). ERV class I SMARCAD1 targets include the VL30 (Virus like 30) and MuLV family, but the most prominent SMARCAD1 bound repeats were class II IAP elements (Fig. 2a). There are an estimated 1000–2000 IAP copies making up about 3% of the mouse genome. LINEs are much more abundant, constituting nearly 20% of the genome[35], yet they displayed only marginal if any SMARCAD1 binding (Fig. 2a). This specificity illustrates that SMARCAD1 accumulation over distinct retroelements is not a consequence of high copy number. MERVL, a class III ERV element primarily repressed by the G9a/GLP complex[15], was not significantly enriched for SMARCAD1 (Fig. 2a). The selective enrichment of SMARCAD1

over particular repeat classes is depicted in a representative genome browser screenshot (Supplementary Figure 3b).

A comparison of the SMARCAD1 profile with the enrichment of H3K9me3 and KAP1 over LTR-retrotransposon categories demonstrated their striking co-occurance over the same elements (Fig. 2a), for example at IAPs (Supplementary Figures 3b, 4c). To determine the extent to which particular transposons are bound by SMARCAD1 and/or KAP1 we performed an association analysis (Fig. 2b and Supplementary Figure 3c). IAP subfamilies as well as a number of other class II and class I elements are enriched for both proteins. Sequential ChIP (Re-ChIP) confirmed that SMARCAD1 and KAP1 are bound simultaneously to classes I and II ERVs (Fig. 2c). The PCR primers recognize consensus sequences and thus amplify the majority of the transposon sub-family members. Additionally, we employed IAP-specific primers designed to detect individual copies, such as the SMARCAD1-bound IAP at the *Mier3* locus (Supplementary Figure 4c).

The ERV subclasses identified as SMARCAD1 targets were verified using a different SMARCAD1 antibody by ChIP-qPCR and the enrichment patterns were found to be phenocopied in two different ESC lines (PGK12.1, Fig. 3a and E14, see below). We observed pronounced enrichment of SMARCAD1 at IAP families with highest occupancy over their 5'UTR (PGK12.1, Fig. 3a, b, top; E14, Supplementary Figure 4d). The SMAR-CAD1 signal at these ERVs was lost after SMARCAD1 depletion (Fig. 3a, b, top). ChIP-qPCR with antibodies against KAP1, H3K9me3, and H4K20me3 showed that SMARCAD1, KAP1 and tri-methylated H3 and H4 are enriched at the same repeat classes (Fig. 3a and Supplementary Figure 4e).

**Lack of SMARCAD1 compromises heterochromatin at ERVs.** To explore how SMARCAD1 occupancy impacts KAP1 binding and histone methylation at retrotransposons, ChIP-qPCR analysis was carried out following SMARCAD1 depletion. While H3 occupancy at TEs was unchanged upon SMARCAD1 removal (Supplementary Figure 4a, b), we observed a decrease of KAP1, H3K9me3, and H4K20me3 levels at IAP elements in stable *Smarcad1* knockdown ESCs (KAP1, H3K9me3: Fig. 3a, b; H4K20me3: Supplementary Figure 4e left). Importantly, these reductions do not reflect long-term adaptation to SMARCAD1 loss as they are also observed upon transient KD (KAP1: Supplementary Figure 5; H3K9me3: Fig. 3c; H4K20me3: Supplementary Figure 4e right). These effects were particularly pronounced at individual IAP elements, such as *Mier3* and *Zfp575* (Fig. 3a and Supplementary Figure 4e). H3K9me3 was additionally affected at several ERV elements other than IAPs

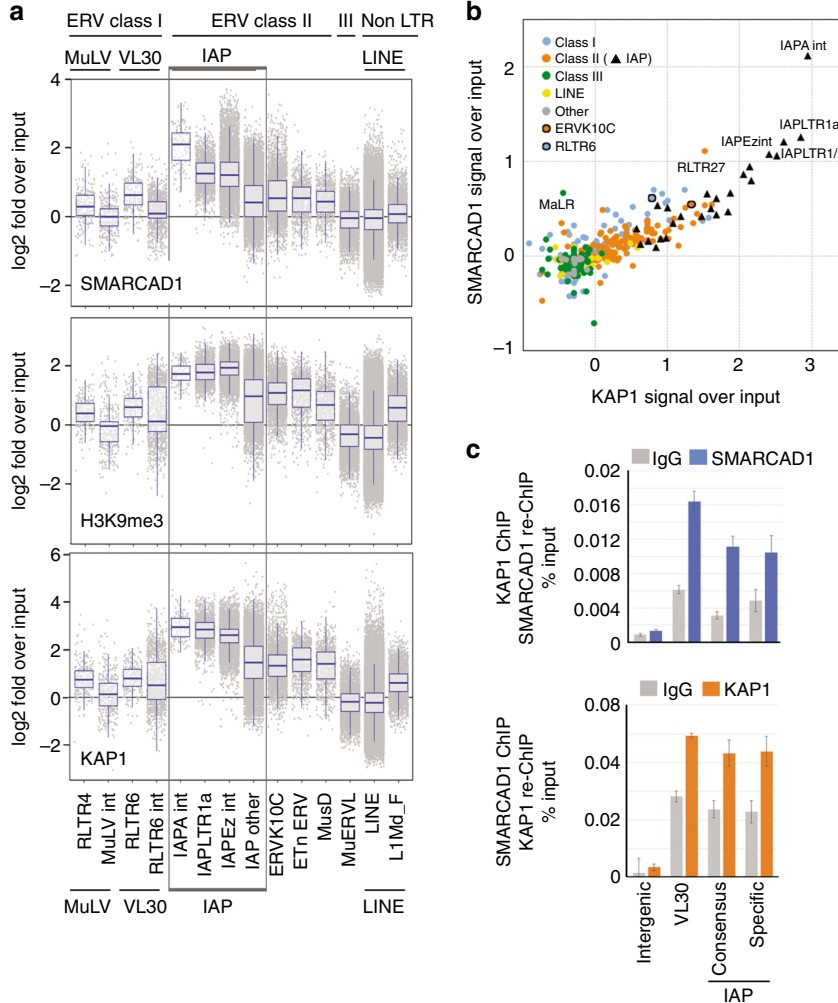

**Fig. 2** SMARCAD1 co-occupies classes I and II ERVs with H3K9me3 and KAP1. **a** Comparison of SMARCAD1 (top, this study), H3K9me3 (middle, this study) and KAP1 (bottom[34]) enrichment at various repeat classes in mESCs. Combined box- and jitterplot depicting log2 fold ratio over input of normalized read counts (TPM) in the indicated ChIP-seq experiments (log2 (TPM ChIP/TPM Input)). In the boxplot center lines show the median; lower and upper box lines correspond to 25th and 75th percentiles, while whiskers extend from the hinge to the smallest (largest) datum not further than 1.5 times the interquartile range. Each dot represents a single repetitive element. Repeat subfamilies were categorized according to their UCSC RepeatMasker annotation. "int" indicates that the repeat is an internal ERV sequence. LTR long terminal repeat. IAP elements are boxed: IAPA_MM-int, IAPEz-int and IAPLTR1a are the most enriched elements, followed by "IAP other" which represents all other IAP elements. LINE: all repetitive elements of the repeat family LINE, except L1Md_F. L1Md_F was analysed separately as it was reported to exhibit high KAP1 enrichment[20]. See also Supplementary Figure 3a. **b** Comparison of SMARCAD1 and KAP1 binding on retrotransposons reveals strong enrichment of both proteins at IAP subfamilies (black triangles). A correlation of binding over several classes I and II ERV family members is apparent. Data points represent median signals over input of SMARCAD1 ChIP-seq data from PGK12.1 ESCs and KAP1 data from ref. [34]. **c** SMARCAD1 and KAP1 co-occupancy at ERV elements of class I (VL30) and class II (IAPs) in mESCs was assessed by sequential re-ChIP using SMARCAD1 and KAP1 antibodies followed by qPCR. Consensus IAP primers recognize the 5′UTR, specific IAP primers an IAP region (*Mier3*) at chromosome 13. An intergenic control site is neither bound by SMARCAD1 nor by KAP1. The percentage of input values are mean ± S.E. of technical triplicates

(Fig. 3a, for example MuLV and MMERVK10C). KAP1 ChIP-Seq in ESCs depleted for SMARCAD1 corroborated that SMARCAD1 loss is accompanied by a reduction of KAP1 occupancy at IAPs (Supplementary Figure 5a, b). ChIP-qPCR analysis following a 2-day depletion of SMARCAD1 moreover showed a subtle, reproducible reduction of KAP1 at other ERV families (Supplementary Figure 5c). These reductions are not the result of a global decrease of KAP1, H3K9me3, or H4K20me3 levels upon SMARCAD1 knockdown (Fig. 1b and Supplementary Figure 4f)[25,27]. Re-expression of SMARCAD1 restored reduced H3K9me3 levels (Fig. 3c, KD+SMARCAD1) and KAP1 binding (see below) observed at IAPs and MMERVK10C elements after the depletion of endogenous SMARCAD1. We thus postulate that

compromised KAP1 and H3K9me3 enrichment at specific ERVs is a direct consequence of diminished SMARCAD1 levels. In conclusion, SMARCAD1 contributes to the establishment and/or maintenance of maximal KAP1 enrichment, H3K9 and H4K20 tri-methylation at a distinctive set of ERVs.

Given that SETDB1 is the primary enzyme responsible for depositing H3K9me3 at classes I/II ERVs[11,13,36], we determined whether the association of SETDB1 with ERV chromatin is altered in SMARCAD1-deficient cells (Fig. 3d). Total levels of SETDB1 were comparable in control (Ctrl) and SMARCAD1 KD cells (Fig. 1b). IAP elements that exhibited reduced levels of H3K9me3 upon SMARCAD1 depletion (Fig. 3a) displayed less SETDB1 binding (Fig. 3d). In contrast, SETDB1 enrichment was

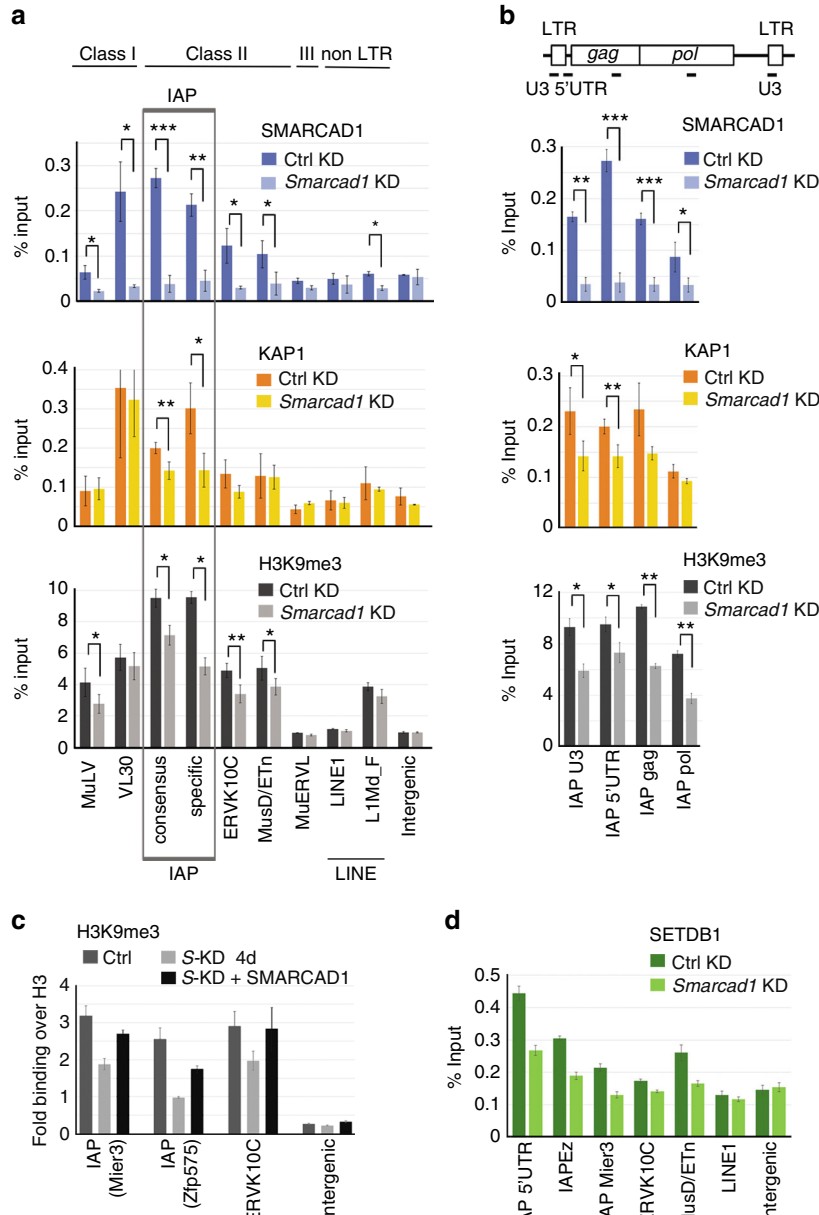

**Fig. 3** SMARCAD1 is required for the maintenance of KAP1, SETDB1, and H3K9me3 at specific ERVs. **a** ESCs depleted for SMARCAD1 and Ctrl knockdown cells were analysed by ChIP with antibodies specific for SMARCAD1, KAP1, H3K9me3, and unmodified H3. Quantitative real-time PCR was conducted using primers specific for indicated retrotransposon classes, confirming their specific enrichment at the same ERV classes I and II. MuERVL elements, as shown previously, were not enriched in KAP1 or H3K9me3[15]. An intergenic region was used as a negative control. Mean enrichment values are presented as percentage of input immunoprecipitated. IAP elements are boxed: the consensus IAP primers shown is for the 5'UTR, the specific IAP primers were designed for the SMARCAD1 bound IAP at the *Mier3* locus (Supplementary Figure 4c). **b** SMARCAD1 is enriched at the 5'UTR of IAPs. Top, schematic of intact IAP structure: LTR, long terminal repeat, *gag*, and *pol* genes. Primer amplicons are indicated as black bars. Below, ChIP-qPCR analysis for SMARCAD1, KAP1 and H3K9me3 over IAP elements in control and *Smarcad1* knockdown cells. SMARCAD1 depleted cells in **b** and **c** show a reduction in KAP1 and H3K9me3 at IAPs. Data are mean ± S.E. from n = 3 (n = 4 ERVK10C) independent experiments in PGK12.1 cells stably depleted for SMARCAD1.
**c** H3K9me3 levels depend on SMARCAD1. ChIP-qPCR data in E14 ESCs showing that SMARCAD1 loss for 4 days results in reduced H3K9me3 levels on ERVs, which are rescued by the expression of exogenous SMARCAD1. Representative target sites and an intergenic control are shown. Real-time qPCR was carried out in technical triplicates and enrichment (±SEM) is the fold change of H3K9me3 over percent input of H3. Western data corresponding to this figure are shown in Supplementary Figure 4f. **d** SMARCAD1 depletion leads to reduced SETDB1 binding to IAPs. SETDB1 ChIP in single-cross-linked chromatin from PGK12.1 cells depleted for SMARCAD1 compared to a control knockdown. qPCR analysis is representative of two biological replicates and presented as mean ± S.E. of technical triplicates. Asterisks denote statistical significance obtained by paired two-tailed Student's *t*-test: *p < 0.05, **p < 0.01, ***p < 0.001

not altered over an intergenic control site. These data suggest that SMARCAD1 facilitates stable occupancy and function of SETDB1 at distinctive ERVs.

Taken together our findings are consistent with a model in which SMARCAD1 is necessary for the stable association of KAP1-SETDB1 complexes on IAP elements and other ERVs to promote H3K9 tri-methylation.

**SMARCAD1 loss disrupts silencing of ERVs and nearby genes**. To test whether SMARCAD1 is required for suppressing transcription of retrotransposons in ESCs we depleted SMARCAD1 by RNA-interference and measured ERV transcript levels using quantitative reverse transcription polymerase chain reaction (RT-qPCR). Elevated transcript levels of SMARCAD1 bound ERV subfamilies were evident 4 days after transfection but not yet at day 2 (Supplementary Figure 6a). For comparison, we conducted a knockdown of *Kap1*, and observed activation of ERVs and nearby genes in a time-dependent manner (Supplementary Figure 6b)[12,16]. KAP1 depleted cells have a propensity to differentiate[12,37,38]. Moreover, SMARCAD1 steady-state levels are progressively diminished upon KAP1 depletion[25]. To minimize the effect of KAP1 depletion on SMARCAD1 levels and cell differentiation, we chose 3 days post transfection for further analysis (Supplementary Figure 6c). For most analysed ERVs the level of upregulation is similar in *Smarcad1* and *Kap1* knockdown ESCs at this time point (Fig. 4a, b). ERVs that behave differently upon KAP1 or SMARCAD1 knockdown include MMERVK10C, where different consensus primers reproducibly measured a higher induction upon KAP1 loss, and elements located within the *Rgs20* gene. We also examined ESCs cultured in the absence of leukemia inhibitory factor (LIF), which leads to differentiation (Fig. 4a, pluripotency marker *Rex1*). Importantly, under these conditions ERVs were not upregulated (Fig. 4b).

Activation of classes I and II ERVs upon SMARCAD1 depletion was confirmed in an additional ESC line (Supplementary Figure 7a) and in ESCs cultured in 2i medium (Supplementary Figure 7b). Consensus ERV primers which detect transcriptional effects occurring across the entire repeat family revealed moderate levels of elevation (~2-fold; Supplementary Figure 7a, b). However, primer pairs directed at specific SMARCAD1 bound ERVs such as an IAP element at the *Bglap3* locus, showed a 3–5-fold increase (Supplementary Figure 7a, b). This indicates that individual ERV copies are differentially responsive to SMARCAD1 depletion and suggests SMARCAD1 controls a subset of ERV elements.

Since transcriptionally reactivated ERVs can influence nearby gene expression[13,22,39,40], we investigated this in *Smarcad1* knockdown cells. We selected mainly genes adjacent to SMARCAD1 bound ERVs previously reported as de-repressed in KAP1-deficient cells as exemplified by *Bglap3* (Fig. 4b)[22,41]. An IAP residing within this gene is bound (Supplementary Figure 8a) and regulated by SMARCAD1 (Fig. 4b, Supplementary Figure 7a, b). De-repression of this IAP upon SMARCAD1 loss was accompanied by a 2–6-fold increase in *Bglap3* mRNA (Fig. 4b bottom panel, Supplementary Figure 7a, b). Likewise, other genes harboring ERVs (*Prnp, Rgs20, Cntnap3)* or located within 1 kb (*Cml2, Zfp575*), 5 kb (*Serinc3*) or 10 kb (*Cyp2b23*) of a SMARCAD1 bound ERV were also dysregulated upon SMARCAD1 depletion (Fig. 4b bottom panel, Supplementary Figure 7a, b). We ruled out that their upregulation is simply a consequence of differentiation (Fig. 4b). Expression of these host genes is hence likely the consequence of de-repression of nearby ERVs. We validated the production of chimaeric transcripts for the *Cyp2b23* and *Cml2* genes by RT-PCR conducted with primers that recognize promoter-proximal ERV elements 5′ to the annotated

gene and an exon (LTR-exon fusion; Supplementary Figures 7a, b and 8a, b).

To distinguish the individual contributions of SMARCAD1 and KAP1 to ERV regulation we examined the dynamics of transcriptional upregulation following removal of either factor (Supplementary Figure 7c). The level of ERV and gene de-repression was broadly similar after 3 and 5 days of SMARCAD1 KD while transcription of the majority of loci tested increased from day 3 to 5 upon KAP1 KD (Supplementary Figure 7c). However, KAP1 depletion for 3 and 5 days was also accompanied by a fall in SMARCAD1 protein levels (Supplementary Figure 7c). Hence, KAP1 knockdown impacts not only on KAP1 function but also diminishes SMARCAD1 levels and thus function. It is therefore not possible to unambiguously determine the extent to which the increase in ERV transcription in KAP1 KD cells is attributable to KAP1 or to a combined SMARCAD1/KAP1 function.

Importantly, restoring SMARCAD1 levels in SMARCAD1 knockdown cells by expressing tagged SMARCAD1 (Supplementary Figure 4f) reversed the upregulation of the *Bglap3* locus and *Cml2* gene (Fig. 4c). We conclude that SMARCAD1 is required for the silencing of distinct ERVs and host genes in their vicinity, thereby safeguarding genome stability in ESCs.

**SMARCAD1 binding to ERVs requires interaction with KAP1**. We next asked how SMARCAD1 is recruited to ERVs. The ESC binding pattern of SMARCAD1 showed a large overlap with KAP1 and H3K9me3 (Fig. 1e, f). To understand whether H3K9me3 impacts on the ERV localization of SMARCAD1 we examined SMARCAD1 occupancy in ESCs depleted for SETDB1 (Fig. 5a). As expected, a significant reduction of H3K9me3 was apparent over ERV classes I/II families in the absence of SETDB1 (Fig. 5a)[11,13]. Under these conditions, binding of SMARCAD1 (Fig. 5a) and KAP1 (Supplementary Figure 9a)[11] to ERVs was not significantly altered. Consequently, neither SETDB1 itself nor tri-methylation of H3K9 are critical for targeting SMARCAD1 to classes I/II retrotransposons. We infer that SMARCAD1 localization to ERVs precedes SETDB1 recruitment.

KAP1 acts upstream of SETDB1 and plays a key role in the recruitment of silencing factors[42]. Therefore, KAP1 constitutes a likely candidate for targeting SMARCAD1 to ERVs. One approach to test this model would be to investigate SMARCAD1 binding to ERVs in *Kap1* knockdown cells. However, since deletion of KAP1 from ESCs reduces SMARCAD1 protein levels[25] we instead investigated the consequence of disrupting the binding of SMARCAD1 to KAP1. FLAG ChIP-qPCR was carried out in previously characterized ESCs that express FLAG tagged SMARCAD1, either wild-type (WT) or a mutant that cannot interact with KAP1[25]. This was achieved with two point mutations in the CUE1 domain of SMARCAD1 (CUE1 mutant F168K, L195K)[25]. We found that SMARCAD1 occupancy at retroelements is significantly reduced in the CUE1 mutant (Fig. 5b and Supplementary Figure 9b). This finding establishes that SMARCAD1 targeting to ERVs is KAP1 dependent.

**Repression of ERVs requires the ATPase function of SMARCAD1**. To determine whether ERV regulation depends on the enzymatic activity of SMARCAD1, we substituted a highly conserved lysine in the ATP binding pocket with an arginine which abrogates ATPase activity (K523R; Fig. 6a). Similar mutations have been employed for the functional interrogation of SWI/SNF type remodelers from yeast to mammals[32,43]. Epitope tagged wild-type (WT) and SMARCAD1 ATPase mutant (mt) vectors were stably introduced into ESCs carrying a doxycycline inducible shRNA targeting the *Smarcad1* 3′-UTR (Fig. 6b). Endogenous

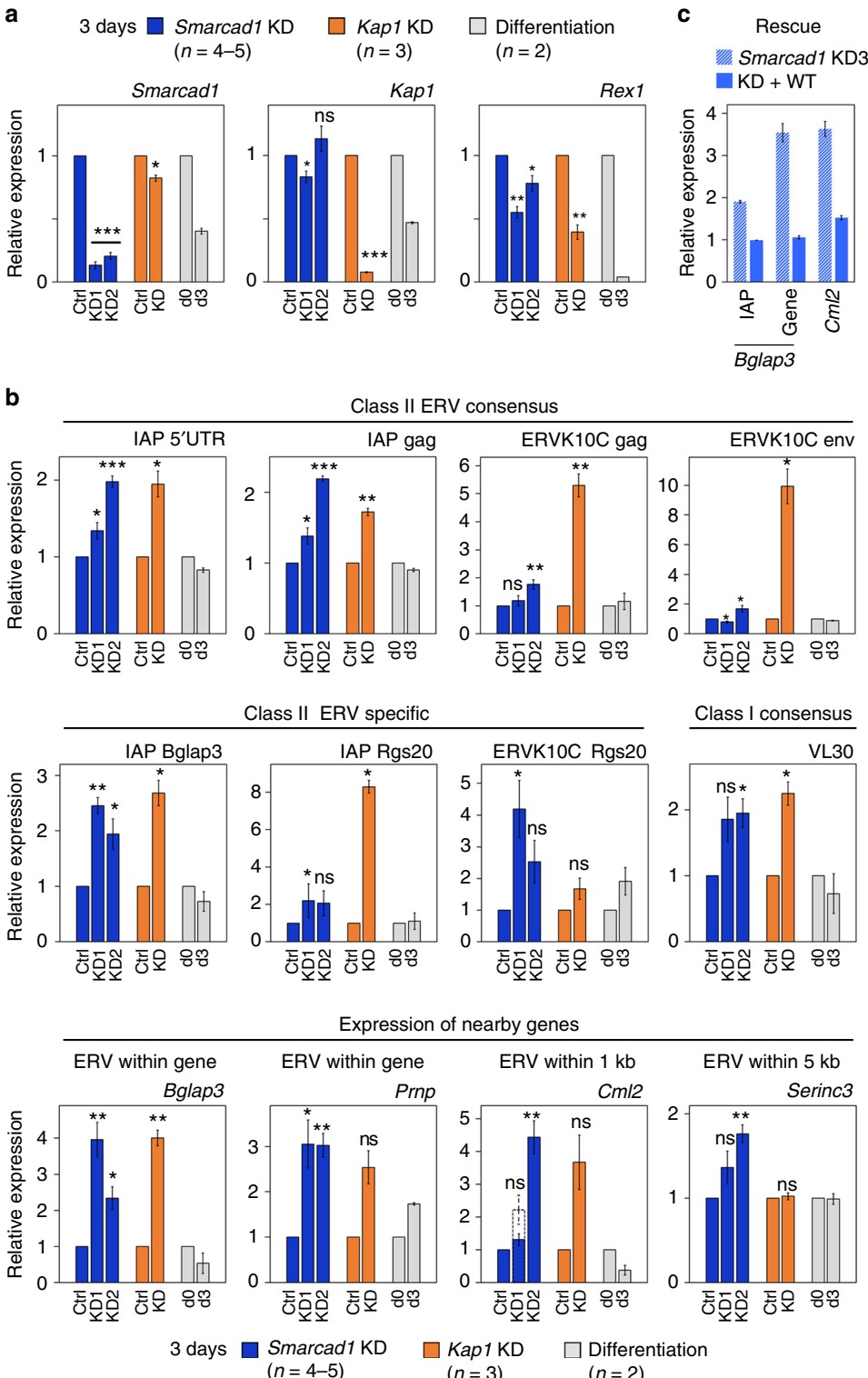

SMARCAD1 protein was subsequently depleted and tagged SMARCAD1 proteins, WT and mutant, were expressed at comparable levels (Fig. 6b, c). Mutating the ATP binding domain did not alter the nuclear localization and cellular fractionation of SMARCAD1 or the protein levels of KAP1 or SETDB1 (Supplementary Figure 10a, b and c).

Earlier, we established that depletion of SMARCAD1 increased the expression of IAPs and the nearby genes *Bglap3* and *Cml2* and

that WT SMARCAD1 was able to restore normal repression at these loci. By contrast, the ATPase mutant SMARCAD1 was unable to effectively repress expression to the levels seen with wild-type protein (Fig. 6d, RT-qPCR). Similarly, the KAP1 binding-deficient SMARCAD1 CUE1 mutant restored repression to some degree but not to the levels seen with the wild-type protein. Furthermore, the reduction of the H3K9me3 mark at IAPs upon SMARCAD1 depletion was not restored by either of

**Fig. 4** Re-activation of ERVs and neighboring genes upon SMARCAD1 knockdown. **a**, **b** RT-qPCR data of SMARCAD1 ($n = 4$–5) or KAP1 ($n = 3$) depleted ESCs (E14), at day 3 after shRNA transfection. Controls include ESC differentiation samples ($n = 2$) derived from cells grown in the absence of LIF for 0 days (d0; undifferentiated) or 3 days (d3, differentiation). Data are the mean ± S.E. normalized to 2–3 housekeeping genes. Similar results were obtained in another ESC line and under 2i conditions (Supplementary Figure 7a, b). **a** Efficiency of each KD was determined with primers specific for *Smarcad1* and *Kap1*. Differentiation was monitored using the pluripotency marker *Rex1*. Protein analysis is shown in Supplementary Figure 6c. **b** Expression analysis of indicated retrotransposons using consensus and specific primers. Bottom row; Expression of *Bglap3, Prnp, Cml2* and *Serinc3* genes, which either harbor SMARCAD1 bound ERVs or are located in the proximity thereof. *Smarcad1* KD1 showed no clear upregulation of *Cml2* at day 3 after transfection, therefore additional analysis on *Cml2* was carried out at day 4 (dotted column; $n = 2$). IGV screenshots depicting analysed loci are in Supplementary Figure 8a. **c** Exogenous SMARCAD1 represses Bglap3 IAP and *Bglap3* and *Cml2* genes (left to right) in *Smarcad1* knockdown cells. RT-qPCR was performed on E14 cells treated with an shRNA against the 3′-UTR of *Smarcad1* for 4 days in the absence (KD3) or presence (+WT) of exogenous SMARCAD1. Error bars present mean ± S.E. of biological duplicates relative to three reference genes. Expression levels of SMARCAD1 were examined by immunoblotting in Supplementary Figure 4f. *P*-values are from paired two-tailed Student's *t*-test: *$p < 0.05$, **$p < 0.01$, ***$p < 0.001$; ns not significant

the mutant transgenes (Fig. 6d, right). In fact, H3K9me3 levels in the mutants were comparable or even lower than those seen in the knockdown. These results show that the ATPase activity of SMARCAD1 and an intact CUE1 domain contribute to the robust transcriptional silencing of ERVs.

We went on to determine the ability of the ATPase mutant SMARCAD1 to bind to ERVs. FLAG ChIP-seq revealed reduced association of the enzymatically inactive SMARCAD1 with IAP elements, compared to the WT protein (Fig. 6e). Likewise, in FLAG ChIP-qPCR experiments the binding of the SMARCAD1 ATPase mutant was reduced over ERVs of classes I and II (VL30; IAPs: 5′UTR, Mier3, Bglap3; ERVK10C; Fig. 6f and Supplementary Figure 10e) and this is also reflected in precipitations with a SMARCAD1 antibody (Supplementary Figure 10d, e). Hence, SMARCAD1 occupancy at ERVs is stabilized by the ability of SMARCAD1 to hydrolyze ATP.

Next, we investigated how the ATPase mutant affects KAP1 binding to ERV chromatin. Importantly, the ATPase mutation does not disrupt the interaction of SMARCAD1 with KAP1 as shown *in vitro* by GST-pulldown (Supplementary Figure 10f) and in cells by endogenous co-immunoprecipitation experiments (Fig. 6g). In ChIP-qPCR analysis, ATPase mutant SMARCAD1 was unable to restore the reduced levels of KAP1 observed at ERVs when SMARCAD1 is depleted. In fact, in these experiments KAP1 levels were even lower than upon SMARCAD1 knockdown (Fig. 6h and Supplementary Figure 10h). In contrast, KAP1 occupancy was restored with WT SMARCAD1 and the KAP1-binding defective mutant (CUE1 mt) that has an intact ATPase domain. Moreover, analysis of KAP1 ChIP-seq data ascertained that maximal KAP1 occupancy at IAPs requires a functional ATPase domain in SMARCAD1 (Supplementary Figure 10g).

These results indicate that the stable association of KAP1 at IAPs and related ERVs is linked to the catalytic activity of SMARCAD1. Collectively, our data suggest that chromatin remodeling by SMARCAD1 is an important factor in the control of ERV elements.

## Discussion

We have elucidated a role for the chromatin remodeler SMAR-CAD1 in the regulation of ERVs in mESC. We show that SMARCAD1 binds specifically to classes I and II ERVs, particularly at IAPs, and present mechanistic insights into how SMARCAD1 prevents their inappropriate activation. A key finding is that the catalytic activity of SMARCAD1 facilitates the binding of KAP1 to ERVs. This underscores a central role for SMARCAD1 in ERV silencing and brings into focus the requirement for ATP-dependent nucleosome remodeling by SWI/SNF-like factors in setting up KAP1-induced heterochromatin formation at ERVs.

KAP1 is known to protect both mouse and human genomes against retrotransposon activity during early development and to

function at exogenous and endogenous retroviruses and at young LINE1 sub-families[20]. We show that KAP1 and SMARCAD1 co-localize at the same ERV subfamilies in mESCs. Neither KAP1 nor SMARCAD1 are known to bind directly to DNA. KAP1 targeting to TEs is mediated by sequence-specific zinc finger proteins[44–47]. Here we demonstrate that the association of SMARCAD1 with ERVs depends on its intact CUE1 domain. KAP1 and SMARCAD1 interact directly via the RBCC domain of KAP1 and the CUE1 domain of SMARCAD1[25]. The tethering of SMARCAD1 to non-viral KAP1 targets such as imprinted control regions also requires the CUE1 domain[25]. Overall, these results point to a general mechanism for localizing SMARCAD1 at different categories of KAP1 binding sites, single-copy genes and ERVs, through the CUE1 motif via its interaction with KAP1. Additional mechanisms targeting SMARCAD1 to chromatin could involve histone modifications since SMARCAD1 interacts with citrullinated H3R26[27].

While KAP1 plays a prominent role in the initial recruitment of SMARCAD1 to retrotransposons, we have generated several lines of evidence that KAP1 occupancy at ERVs itself is influenced directly by SMARCAD1. First, SMARCAD1 knockdown leads to reduced KAP1 occupancy at IAPs and other ERV families, suggesting that the initial recruitment and/or the retention of KAP1 is affected. Second, introducing a point mutation in the ATP binding pocket of SMARCAD1 results in reduced SMARCAD1 enrichment at ERVs and interferes with KAP1 binding. Therefore, ablation of the ATPase activity mimics the complete removal of SMARCAD1. This indicates that active chromatin remodeling maintains normal levels of SMARCAD1 and KAP1 at IAPs. Together, our data reveal that SMARCAD1 and KAP1 are functional, co-dependent partners in the regulation of ERVs.

KAP1 recruitment to specific ERVs is essential for their silencing since KAP1 serves as a scaffold for heterochromatin inducing factors, including SETDB1[6,9]. We found that SETDB1 levels were reduced at the two most active ERV families, IAPs and MusD/ETn, upon SMARCAD1 loss. Conversely, SMARCAD1 binding to ERVs was not disrupted upon SETDB1 depletion. Our results therefore place SMARCAD1 early in the sequence of events leading to full ERV repression, coincident with KAP1 and upstream of SETDB1.

The functional consequence of SMARCAD1 depletion is re-activation of ERVs, indicating that the underlying chromatin structure has changed to a less repressive state. Consistent with this we observed reduced levels of repressive histone modifications, H3K9me3 and H4K20me3, at ERV subfamilies upon *Smarcad1* knockdown. H3K9me3 levels at IAPs and MMERVK10C were restored upon re-expression of wild-type SMARCAD1. In direct support for a requirement of SMARCAD1 function in the maintenance of maximal H3K9me3 levels at these elements, a catalytically inactive SMARCAD1 failed to rescue

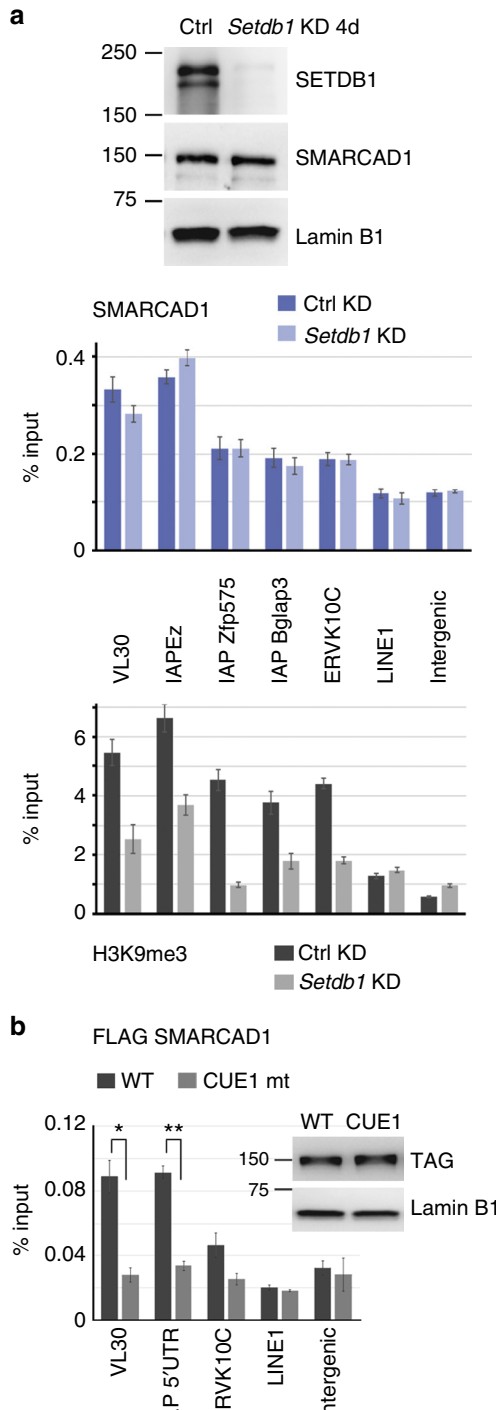

**Fig. 5** SMARCAD1 binding to ERVs is not dependent on SETDB1 but on KAP1. **a** Western blot analysis in E14 mESCs transfected with a *Setdb1* shRNA (KD) or a non-targeting control (Ctrl) 96 h post transfection shows efficient knockdown. SMARCAD1 levels were not affected. Lamin B1 serves as loading control. Single-cross-linked chromatin was prepared from these cells and ChIP was carried out with H3, H3K9me3, SMARCAD1, and KAP1-specific antibodies. qPCR over retrotransposons reveals a clear reduction of H3K9me3 but not of SMARCAD1 over class I (VL30) and class II (IAPs, MMERVK10C) ERVs upon *Setdb1* knockdown. Data are representative of two immunoprecipitations and the error bars denote the mean ± S.E. of technical triplicates. Corresponding H3 and KAP1 ChIPs are shown in Supplementary Figure 9a. **b** Stable association of SMARCAD1 with ERV subfamilies depends on an intact CUE1 domain in SMARCAD1. ChIP-qPCR analysis in ESCs depleted for 2 days of endogenous SMARCAD1 protein but expressing 3X FLAG tagged SMARCAD1, either WT or a mutant that affects its interaction with KAP1 (CUE1 mt, F168K, L195K[25]). Western blot shows that tagged WT and CUE1 mutant SMARCAD1 proteins are expressed at similar levels in the cells utilized for ChIP. ChIP was carried out with a FLAG-specific antibody on double-cross-linked chromatin and analysed by qPCR. Depicted is the mean ± S.E. from three biological replicates (n = 3). P-values are from paired two-tailed Student's t-test: *p < 0.05, **p < 0.01. Additional analysis with specific IAP primers is shown in Supplementary Figure 9b

functional interdependence of SMARCAD1 and KAP1 in ERV regulation. Mutations in SMARCAD1 that abolish its interaction with KAP1 could not silence transcription of IAPs and nearby genes as effectively as WT SMARCAD1 and displayed reduced H3K9me3 levels over IAPs. We hypothesize that under normal conditions the interaction of SMARCAD1 with KAP1 localizes the ATPase activity of SMARCAD1 to IAP chromatin; but in the absence of this interaction the ATPase activity is not readily available, ultimately resulting in compromised silencing. In support, we find that abolishing the ATPase function of SMARCAD1 disrupts the efficiency of KAP1-SETDB1-mediated IAP silencing. Our findings have uncovered another layer of regulatory control of ERV silencing and present a framework in which KAP1-SMARCAD1 function can be dissected elsewhere in the genome.

How might a functional ATPase domain in SMARCAD1 contribute to ERV repression? SMARCAD1 orthologs exhibit nucleosome sliding and histone exchange activities in vitro[48,49]. Fission yeast SMARCAD1 subfamily members have been reported to regulate a gypsy class LTR retrotransposon via transcription start site selection by modulation of nucleosome occupancy at LTR elements[50]. SMARCAD1 activity could similarly affect the chromatin structure of the target region and/or modulate the function of ERV binding proteins such as sequence-specific zinc finger proteins which recruit KAP1 to ERVs[44–47]. Effects on chromatin structure might involve changes in the position or composition of nucleosomes but also in post-translational modifications. One candidate is sumoylation, which enhances the recruitment of KAP1 and is required for its repressive function[9,51,52]. SMARCAD1 might facilitate the access of SUMO conjugating enzymes to KAP1-bound loci. A role in providing efficient access to histone modifiers has been previously ascribed to the nucleosome remodeling and deacetylation (NuRD) complex[53]. Since both SMARCAD1 and KAP1 have acknowledged functions in the re-establishment of heterochromatin following DNA replication[43,54], an intriguing possibility is that remodeling helps to propagate ERV silencing through cell division.

It has recently become apparent that the KAP1-SETDB1 system contributes to the somatic control of TEs[41,55–57]. As SMARCAD1 is expressed widely and its interaction with KAP1 is not restricted to mESCs[43,54,58], an involvement of SMARCAD1 remodeling in the regulation of ERVs in different cell types and

H3K9me3 levels. We further demonstrate that SETDB1 co-immunoprecipitates with SMARCAD1, as does HDAC1. The model emerging from our data is that SMARCAD1 binding at IAPs together with KAP1 promotes the recruitment of SETDB1 and other chromatin modifiers which create a repressive chromatin environment, thereby preventing inappropriate ERV activation (Fig. 7).

Our expression studies showed that SMARCAD1 depletion not only has an effect on ERVs, but also results in misregulation of nearby cellular genes. To achieve maximum levels of ERV de-repression, KAP1 KD is needed, further emphasizing the

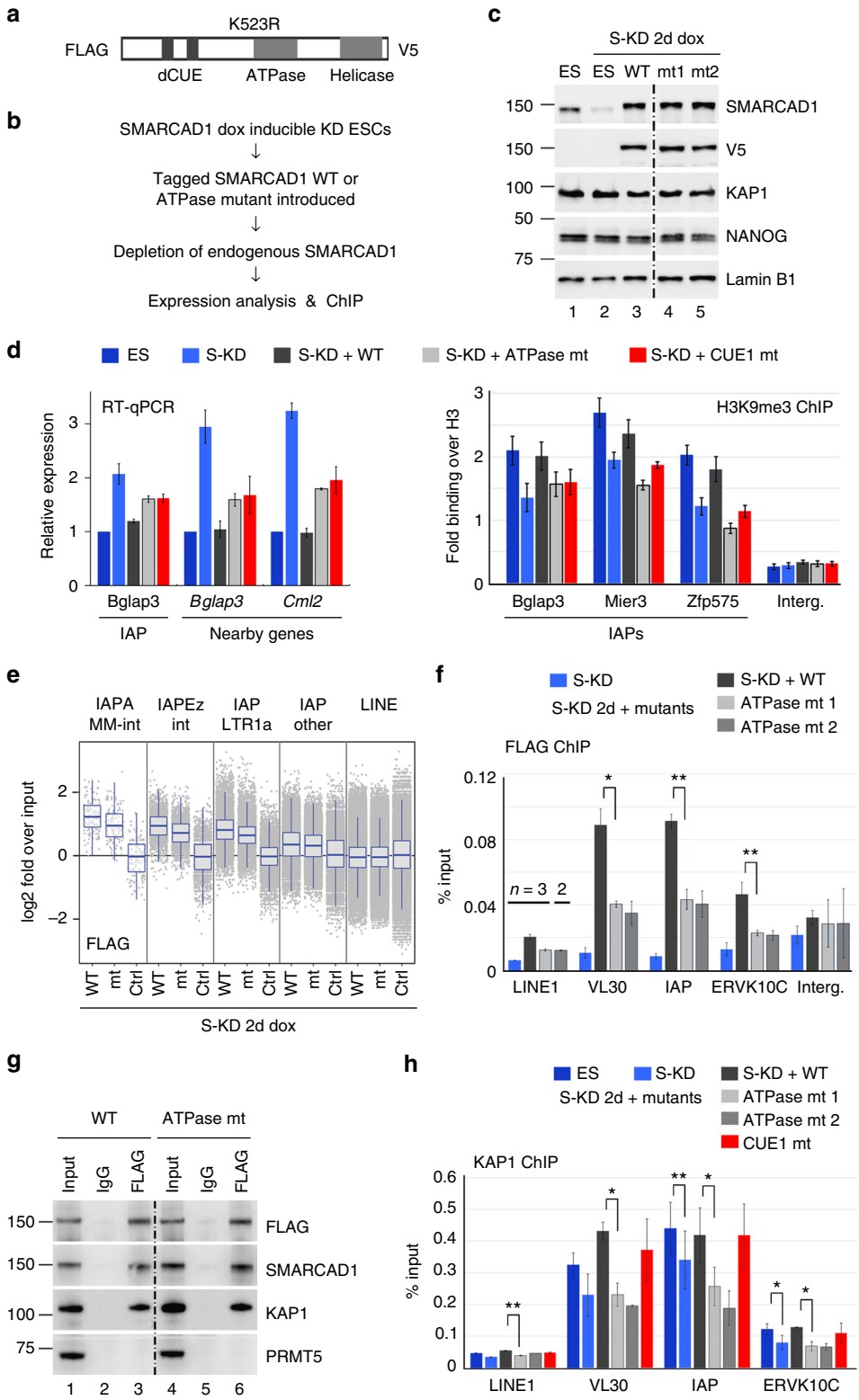

developmental stages is likely. Our ChIP-seq analysis suggests that the association of SMARCAD1 with KAP1 is not limited to ERVs but extends to other KAP1 targets. KAP1 is required for the maintenance of genomic imprints, which, akin to ERV regulation, involves heterochromatin formation[59]. In a separate study we confirmed SMARCAD1 binding to imprinted genes, supporting a possible role for SMARCAD1 in their regulation[25]. We thus postulate that the partnership of SMARCAD1 and KAP1 in the establishment/maintenance of heterochromatin as described here may have broader significance beyond retrotransposon biology.

**Fig. 6** An active ATPase domain is required for SMARCAD1 function at ERVs. **a** Schematic of mSMARCAD1 showing the double CUE (dCUE), the ATPase/helicase domains and the K523R mutation. Location of the FLAG and V5 tags are indicated. **b** Workflow: E14 cells carrying an inducible *Smarcad1* shRNA construct were stably transfected with FLAG-SMARCAD1-V5 constructs. Endogenous SMARCAD1 was depleted before RT-qPCR and ChIP was performed. **c** Characterization of E14 ESCs expressing tagged SMARCAD1 constructs by western blot. Inducible SMARCAD1 knockdown E14 ESCs (lane 1) show depletion of SMARCAD1 after 2 day doxycycline treatment (dox; lane 2). Cells expressing WT (lane 3) and mutant SMARCAD1 (mt1 and mt2, lanes 4–5, K523R) were monitored using an anti-SMARCAD1 antibody, detecting both endogenous and tagged protein, and an anti-V5 antibody, for tagged SMARCAD1. Lamin B1 serves as a loading control. Dotted line indicates discontinuous lanes from the same gel. **d** RT-qPCR and ChIP-qPCR analysis in SMARCAD1 knockdown cells (S-KD) rescued with SMARCAD1 transgenes; wild-type (WT), K563R (ATPase mt) and F168K, L195K (CUE1 mt). Left panel; RNA was collected 5 days after depletion of endogenous SMARCAD1 to investigate expression of IAPs and nearby genes. Data confirming KD and similar levels of FLAG SMARCAD1 proteins between cell lines are in Supplementary Figure 10c. Relative expression is mean ± S.E. of two biological replicates normalized to three housekeeping genes. Right panel; H3 and H3K9me3 ChIP-qPCR on double-cross-linked chromatin before and 4 days after SMARCAD1 depletion. qPCR was carried out in triplicates and fold binding of H3K9me3 over H3 (±S.E.) is presented. **e** SMARCAD1 WT and ATPase mutant binding at IAP elements. FLAG ChIP-seq following depletion of endogenous SMARCAD1 in ESCs expressing FLAG tagged SMARCAD1 wild-type (WT) or ATPase mutant (mt) or in ESCs lacking FLAG proteins (Ctrl). For each repeat element the log2 fold ratio over input was plotted as in Fig. 2a, Box lines show the median, 25th and 75th percentiles; whiskers end at the smallest (largest) datum not further than 1.5 times the interquartile range. LINE elements show no enrichment. **f** ChIP of double-cross-linked chromatin from the cells described in **b**, **c** with a FLAG antibody, which detects tagged SMARCAD1 WT and ATPase mutant. qPCR analysis over representative ERV elements of class I (VL30) and class II (IAPs and MMERVK10C) is shown. Analysis of additional IAP elements and SMARCAD1 ChIP is shown in Supplementary Figure 10d, e. **g** The ATPase mutation does not disrupt the association of SMARCAD1 with KAP1. A FLAG antibody co-immunoprecipitates KAP1 from ESCs expressing FLAG tagged SMARCAD1 protein, both WT (lane 3) or the ATPase mutant K523R (lane 6) in the presence of ethidium bromide and benzonase. Lanes 1 and 4, 3% input. PRMT5 served as a negative control. Endogenous SMARCAD1 was depleted by 2 day doxycycline treatment. Dotted line indicates discontinuous lanes from the same gel. **h** SMARCAD ATPase function facilitates stable KAP1 occupancy at ERVs. Binding of KAP1 was analysed by ChIP-qPCR over the same sites and in the same cells as in **f** and Fig. 5b (CUE1 mutant). Additional sites are in Supplementary Figure 10h. qPCR data in **f** and **h** are shown as mean ± S.E. of three-independent experiments (except ATPase mt 2 (*n* = 2) which precludes it from statistical analysis). *P*-values were calculated using paired two-tailed Student's *t*-test: *\*p* < 0.05, *\*\*p* < 0.01

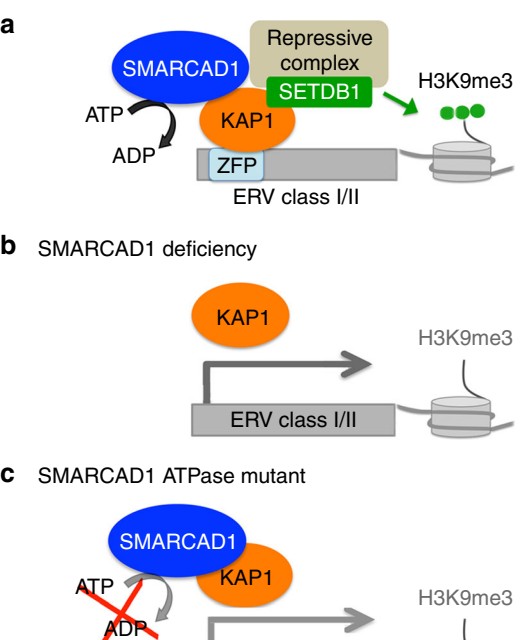

**Fig. 7** Model; SMARCAD1 function in the KAP1-SETDB1 ERV silencing pathway. **a** ERVs recruit DNA binding zinc finger proteins (ZFP), KAP1, SMARCAD1 and repressor proteins, such as SETDB1, which induce heterochromatin formation, leading to transcriptional silencing. The catalytic activity of SMARCAD1 facilitates stable KAP1-SMARCAD1 binding at ERVs. **b** In the absence of SMARCAD1, KAP1 protein levels are not affected, but its binding to ERVs and the recruitment of SETDB1 are compromised, leading to less H3K9me3 and induction of ERV expression which spreads to neighboring genes. **c** A mutation in the ATPase domain of SMARCAD1 does not disrupt the physical interaction with KAP1, but interferes with stable SMARCAD1 and KAP1 binding to ERVs, resulting in inefficient H3K9 tri-methylation and inefficient silencing

## Methods

**Cell lines, plasmids and transfections**. MEFs and primary ear fibroblasts were cultured in DMEM (Invitrogen) supplemented with 10% FBS and L-glutamine. Feeder free mouse ESCs (XY: E14, kind gift from Dr. Haruhiko Koseki; J1, ATCC SCRC-1010; XX: PGK12.1, generous gift from Dr. Neil Brockdorff) were grown on gelatin-coated surfaces in standard ESC medium (DMEM supplemented with 15% FBS (fetal bovine serum), 0.1 mM 2-mercaptoethanol, 0.1 mM non-essential amino acids, 2 mM L-glutamine supplemented with 1000 U/ml LIF). 2i-culturing conditions were established by culturing E14 ESCs in N2B27 medium supplemented with MAPK and GSK inhibitors (3 μM and 0.6 μM, respectively) and LIF[60]. N2B27 medium was prepared by mixing equal volumes of DMEM/F12 (Invitrogen) and Neurobasal Medium (Invitrogen), and supplemented with 0.5X B-27 supplement (Invitrogen), 0.5X N-2 supplement solution (Invitrogen), 1 mM L-glutamine, and 0.1 mM β-mercaptoethanol. Doxycycline was used at a final concentration of 0.5 μg/ml. Cell lines were routinely tested for mycoplasma infection using MyoAlert Detection (Lonza). Differentiation of ESCs was achieved by removal of LIF[61]. Transfection of plasmids into ESCs was performed using Lipofectamine 2000 (Invitrogen) according to the manufacturer's instructions. Detection of alkaline phosphatase, a marker of the undifferentiated ES cell state, was carried out using the Millipore (SCR004) kit.

Generation of cell lines expressing tagged SMARCAD1: The coding sequence of SMARCAD1 was amplified from mouse ESC cDNA and inserted into a chicken β-actin promoter (CAG) -driven expression vector, with an N-terminus triple FLAG tag. Tagged wild-type *Smarcad1* construct along with the empty FLAG vector control was transfected into PGK12.1 ES cells, followed by 1.7 μg/ml puromycin selection and clonal expansion. The ATPase mutation K523R was generated by site-directed mutagenesis of mouse *Smarcad1* using the QuikChange Lighting Site Directed Mutagenesis Kit (Agilent 210518) and the following oligos:

Fwd 5'-GCAGACGAAATGGGCCTAGGAAGAACCATTCAAGCCATTG
C-3'

Rev 5'-GCAATGGCTTGAATGGTTCTTCCTAGGCCCATTTCGTCTGC-3'

3X FLAG tagged *Smarcad1* wild-type and ATPase mutant constructs were additionally tagged at the C terminus with a V5 tag and stably integrated into E14 ESCs carrying a doxycycline inducible shRNA targeting the 3'-UTR of *Smarcad1* as described[25] prior to knocking down the endogenous *Smarcad1*. Cells with stable integrants were selected using 1 μg/ml puromycin. Clonal cell lines (ATPase mutants 1 and 2) or pools (wild-type) representing a number of independent cell clones were analysed by indirect immunofluorescence and western blot.

**Knockdown (KD) experiments by RNAi**. Transient depletion of KAP1 was accomplished with pLKO.1 puro shRNA vectors. The shRNA sequence targeting Kap1 is 5'-TTGAACTGTTTGAACATGC-3'[25]. An shRNA does not target any mouse gene was used as negative control: 5'-CGAGGGCGACTTAACCTTAGG-3' For Setdb1 knockdown E14 cells were transfected twice, the second transfection two days after the first one, with a pLKO.1 based shRNA construct: 5'-

CCCGAGGCTTTGCTCTTAAAT-3'[60]. Cells were collected 4 days after the first transfection.

Depletion of SMARCAD1 from ESCs was achieved both transiently and by generation of stable knockdown ESC lines. We used a doxycycline inducible system[25] or shRNA vectors encoding SMARCAD1-specific hairpins in pSUPERpuro, pHYPER or CAG-eGFP-miRE-IRES-Puro plasmid backbones previously described[25,28,62]. Specifically, the sequences target either exons or the 3'-UTR:

shRNA Smarcad1 Exon 7: 5'-GGACTATAGCAGTTGTGAA-3' in pHYPER
shRNAs Smarcad1 Exon 12: 5'-GTATGAGGATTACAATGTA-3' and
5'-GAAGAGCGTAAGCAAATTA-3' in pSUPER
shRNA Smarcad1 3'-UTR: 5'-TTAAGTTAATCTGTTCTGCTGG-3'

shRNA vectors encoding non-target controls such as luciferase or linker sequences that do not form a hairpin were employed in parallel[25,63,64]. Knockdown efficiency was determined by a combination of western blotting, indirect immunofluorescence or quantitative reverse transcription PCR.

**Chromatin immunoprecipitation**. Chromatin immunoprecipitation was performed using the One Day ChIP kit (Diagenode C01010080) after either a single-cross link with 1% formaldehyde or double-cross-linking. For the latter, ~$5 \times 10^7$ ESCs per 15 cm plate were cross-linked with 2 mM DSG (disuccinimidyl glutarate, Thermo Scientific 20593) in PBS for 45 min at room temperature (r.t.), washed three times with PBS, followed by cross-linking with 1% formaldehyde in PBS (Polysciences 04018) for 10 min at r.t. Formaldehyde was quenched using 125 mM glycine. After cell lysis, chromatin was sheared in a Bioruptor (Diagenode) to produce fragments of ~200–600 bp.

Immunoprecipitations were performed following the protocol provided by the manufacturer using 100 μg of DNA and routinely 3 μg of antibody (antibody list see Supplementary Table 2). ChIP DNA was analyzed in triplicate using real-time PCR with Sybr Green (Bio-Rad) on a CFX96 Connect (Bio-Rad) or Agilent MX3000P. Enrichment values are expressed routinely as percentage of input or, for histone modifications, as fold change over H3 using error propagation. Primer sets used for qPCR are available in Supplementary Table 3. qPCR conditions were as follows: 10 min at 95 °C followed by 40 cycles at 95 °C for 15 s and 60 °C for 30 s, followed by a plate read after each cycle. Melting curve test was performed at the end of each experiment (from 55 to 95 °C, read plate every 0.5 °C) to ensure the specificity of amplification.

**ChIP-sequencing**. For ChIP-seq, chromatin was eluted in 1% SDS, 100 mM NaHCO3, incubated with NaCl o/n, treated with proteinase K and purified by QIAquick columns (Qiagen). Library preparation was performed using the MicroPlex Library Preparation Kit v2 (Diagenode) according to manufacturer's instructions. Briefly: 4–8 ng DNA was used for each sample. Adapter-ligated DNA was subject to 9 cycles of PCR amplification before size selection and DNA purification with AMPure XP beads (Agencourt). Size and concentration were assessed on a Bioanalyzer (Agilent Technologies). Sequencing was performed on the Illumina 1500 platform with on-board cluster generation using the HiSeq Rapid SR Cluster Kit v2 (Illumina) and single read 50 nucleotide sequencing on a HiSeq Rapid SR Flow Cell v2 (Illumina).

**Re-ChIP**. Sequential ChIP was performed using in a first ChIP step a SMARCAD1 (3 μg) or a KAP1 (3 μg) antibody. The precipitated material was eluted twice from the beads with 10 mM DTT, 100 mM NaCl, and 1% SDS at 37 °C for 30 min. The eluted DNA was diluted 40-fold with ChIP buffer and subjected to a second round of immunoprecipitations in accordance with the One Day ChIP kit (Diagenode) manual, carrying out an overnight antibody (6 μg) incubation at 4 °C with KAP1, SMARCAD1 and isotype control IgG antibodies. Real-time qPCR was executed and enrichment values were calculated relative to the input of the first ChIP.

**RNA extraction and expression analysis**. Total RNA was extracted from ESCs using TRIzol (Invitrogen, 15596026) followed by DNase digest using TURBO DNA-free™ Kit (Ambion, AM1907). Routinely, cDNA was synthesized from 0.8 μg of total RNA with random hexamers (Invitrogen, N8080127) using SuperScript II reverse transcriptase (Invitrogen, 18064014). Reactions without reverse transcriptase were processed in parallel to control for genomic DNA background. qPCRs were performed using a CFX Connect Real-Time PCR Detection System (Bio-Rad) or Agilent MX3000P, using iTaq Universal SYBR Green (Bio-Rad). Pairs of primers were evaluated for generating single-peak melting profiles and for linear amplification over a range of DNA template dilutions. qPCR assays were performed in triplicates. Samples were normalized to housekeeping genes (Gapdh, Hsp90ab1, Atp5b) and expression levels were calculated with the Bio-Rad CFX Manager software (version 3.1), which uses a ΔΔCq calculation scheme. In rescue experiments of Smarcad1 knockdown cells with Smarcad1 transgenes the expression differences were calculated in relation to the untreated parental cells. One exception is Cml2 since treatment with selection agent (puromycin) resulted in increased expression. Therefore these values were subtracted to generate the appropriate baseline. Primers are listed in Supplementary Table 3. The expression heatmap was generated in Excel (Microsoft) using the function of Conditional Formatting on the log2 fold changes.

**Extract, co-immunoprecipitations, and western blots**. Nuclear extracts of mESCs were made by salt extracting isolated nuclei. Scraped cells were pooled and pelleted at 1250 g for 5 min; the measured packed cell volume (PCV) was used as reference volume for subsequent steps. The cells were resuspended in 2X PCV of Buffer A (10 mM HEPES, pH 7.6, 1.5 mM MgCl2, 10 mM KCl) and homogenized in a Dounce homogenizer using a loose pestle. After centrifugation nuclei were resuspended in 1.5X PCV of Buffer C (20 mM HEPES, pH 7.6, 0.2 mM EDTA, 1.5 mM MgCl2, 420 mM NaCl, 20% glycerol) and homogenized using a Dounce homogenizer with a tight pestle. The homogenate was incubated in the cold room for 30 min, centrifuged to separate the supernatant and dialyzed against Buffer D (20 mM HEPES, pH 7.6, 0.2 mM EDTA, 1.5 mM MgCl2, 100 mM KCl, 20% glycerol).

For total protein extraction, cell pellets were re-suspended in 3X pellet volume 20 mM HEPES, pH 7.3, 110 mM KOAc, 5 mM NaOAc, 2 mM MgOAc, 1 mM EGTA, 2 mM DTT, 0.1% NP-40, 10 mM MnCl2 supplemented with 20 μg/mL DNase I, incubated at 37 °C for 30 min, with occasional inverting to disperse precipitates. For co-immunoprecipitation, the lysate was centrifuged (13,000×g for 5 min at 4 °C) to remove insoluble fractions. For western blot analysis, the lysate was boiled in 1X Laemmli buffer.

Fractionations into soluble and chromatin fractions: ESCs were trypsinized and washed in ice-cold PBS supplemented with 1 mM benzamidine and 0.5 mM PMSF. Cell pellets were resuspended in 1X packed cell volume (PCV) CSK buffer (10 mM PIPES, pH 6.8, 300 mM sucrose, 100 mM NaCl, 3 mM MgCl2, 1 mM benzamidine, 0.5 mM PMSF) with 0.1% Triton X-100 and incubated on ice for 3 min. The lysate was centrifuged at 12,000 rpm for 5 min at 4 °C. The supernatant was collected as the soluble fraction. The pellet was washed once briefly with 1X PCV CSK buffer and then resuspended in 1X PCV CSK buffer supplemented with 300 units/ml Benzonase. After 10 min on ice, the reaction was stopped with 5 mM EDTA. Extract amounts corresponding to equal number of cells were analyzed by SDS-PAGE and immunoblotting.

Co-immunoprecipitation experiments were carried out as follows[25]: precleared extracts, either 150 μg of nuclear extract or 300 μg of whole-cell extract, were incubated with 3–3.5 μg of specific antibody or IgG. A total of 150 units/ml Benzonase (Novagen) and 0.1 μg/μl EtBr were added during the experiment to decrease interactions facilitated by nucleic acid. Immune complexes were captured by Protein G Dynabeads (Novex). Dynabeads-Ab-Ag complexes were washed four times in 20 mM HEPES, pH 7.6, 100 mM KCl, 0.2 mM EDTA, 1.5 mM MgCl2, 0.5 mM DTT, 20% glycerol, 0.05% Triton X-100 with added protease inhibitors and then washed once in an identical buffer but containing 50 mM NaCl. Immune complexes were eluted with 1X Laemmli buffer.

For western blotting, samples in 1X laemmli buffer were separated by SDS-PAGE and transferred to nitrocellulose or PVDF membranes. Membranes were blocked in PBST buffer (PBS, 0.1% Tween 20, 5% w/v low-fat dry milk) for 1 h at room temperature and incubated with primary antibodies listed in Supplementary Table 2 overnight at 4 °C. Membranes were washed three times for 10 min with PBST, incubated for 1 h with secondary antibody conjugated to horseradish peroxidase (HRP) and washed three times for 10 min with PBST and 10 min with PBST without milk. Detection was performed using Immobilon™ Western chemiluminiscent HRP substrate (Millipore), images were captured with X-ray film or digitally using the ChemiDoc (Bio-Rad) imaging system.

**Superose 6 gel filtration**. Size fractionation of protein complexes was carried out on an AKTA FPLC. PGK12.1 ESC nuclear extract (500 μl; 1.6 μg/μl) was dialyzed into 10 mM HEPES, pH 7.6, 0.5 mM EGTA, 1.5 mM MgCl2, 300 mM KCl, 10% glycerol and separated on a Superose 6 gel filtration column (HR 10/30, GE Healthcare) in the same buffer. Elution fractions (0.5 ml) were TCA (trichloroacetic acid) precipitated, subjected to SDS-PAGE and analyzed by western immunoblotting with specific antibodies. The Superose 6 column was calibrated with gel filtration calibration standards (GE Healthcare).

**GST pull down assays**. Glutathione-S-transferase (GST) and human KAP1ΔPB (AA 1-628) GST fusion proteins in a pGEX-4T-1 vector[25] were expressed in E. coli and purified using a C3 Liquidiser (Avestin Europe GmbH). Recombinant V5-human SMARCAD1 proteins, either WT or an ATPase domain mutation (K523R), were generated in a T7 TNT reaction (Promega). Binding reactions using GST fusion proteins and target proteins produced in the T7 TNT reaction were performed in 20 mM HEPES, pH 7.5, 75 mM KCl, 0.1 mM EDTA, 2.5 mM MgCl2, 1 mM DTT, 0.05% Triton X-100 as recommended by the supplier[25]. Beads were washed three times in binding buffer, and bound proteins were eluted from the beads with two sequential extractions in SDS sample buffer at room temperature. Bound proteins were identified by western blotting.

**Flow cytometry**. For FACS ESCs were resuspended in 200 μl of PBS, fixed by dropwise addition of 1.3 ml of ice-cold 70% ethanol, incubated at 4 °C for at least 1 h. Cells were collected by centrifugation and incubated in PI/RNase staining buffer (BD Pharmingen, 550825) for 15 min at room temperature. Analysis was performed on a BD LSR II Cytometer and with FlowJo software (version 10.2.).

**Immunofluorescence**. For indirect immunofluorescence cultured cells were dropped on gelatin-coated glass slides prior to fixation with 4% formaldehyde for 5 min (FLAG), 10 min (SMARCAD1), or 15 min (V5). Permeabilization was performed for 10–15 min in PBS/Triton X-100, namely 0.1% for SMARCAD1, 0.2% for V5, and 0.4% for FLAG. Counterstaining of nuclei was carried out with mounting medium with DAPI (Vector Laboratories, H-1200). Where specified, pre-extraction was performed with 0.5% Triton X-100 in CSK buffer (10 mM PIPES, pH 6.8, 300 mM sucrose, 100 mM NaCl, 3 mM $MgCl_2$) for 35 s on ice in the presence of 0.5 mM PMSF followed by washes in ice-cold PBS and incubation in 4% formaldehyde (Sigma 47608) for 10 min. For direct comparison, different cell lines were grown on the same slide and stained and processed together; images were acquired using the same exposure time on a LEICA DMR fluorescent microscope, and identical post-processing with Adobe Photoshop CS3 or FIJI was applied.

**Proliferation, IncuCyte imaging and eccentricity assays**. Growth curves for PGK12.1 and E14 ESCs were carried out by seeding $1–2 \times 10^4$ Ctrl and *Smarcad1* knockdown cells in triplicate into 24-well plates. Cells were counted in 24 h time intervals using a CASY Cell Counter. For observing cell growth in real time cells were plated in triplicates at dilutions ranging from 10,000 cells to 312 cells per well on 96-well tissue culture plates. Photomicrographs were captured every 3 h using an IncuCyte cell live imager (Essen BioScience) and eccentricity of the cultures was measured using the software supplied by IncuCyte.

**ChIP-Seq data analysis**. Reads were aligned to the *Mus musculus* genome retrieved from Ensembl revision 83 (mm10) using Bowtie 2.0.0-beta7[65], using the default parameter settings. After alignment the lanes were de-duplicated to the expected number of duplicate reads based on binomial distribution, keeping only these effective reads for further analysis.

Peak calling was performed individually for each sample using antibody control or input as background. The MACS program v1.4.0rc2[66] was used for all samples, except for the histone modification mark H3K9me3. H3K9me3 ChIP peaks were called using SICER 1.1[67] with these parameters: windows size 200, gap size 200, fragment size 51, mappability percentage 0.78. For samples with corresponding antibody and input backgrounds only those peaks were kept that were called against both backgrounds.

Peak filtering was used to identify only those peaks showing a strong enrichment over their background(s) and thereby reduce false-positive background. SMARCAD1 and KAP1 peaks were only retained if they had a minimum of 30 effective foreground reads, not more than 50 effective reads in either of their background(s), and showing at least a 2.5-fold increase in the normalized read counts (TPM) compared to their background(s). Only H3K9me3 peaks showing at least a threefold increase in the normalized read counts compared to either of their backgrounds were kept. To enable comparison between the samples, tag counts were calculated and normalized to one million mapped reads (TPM, tag per million). The foreground-background ratio used to filter reported peaks was calculated on basis of TPMs in foreground versus TPMs in background. If more than one background value was available the ratio was calculated using the maximum background TPM value.

Average Signal plot: All reported peaks from endogenous SMARCAD1-wild-type and SMARCAD1-KD ChIP-Seq were used to plot averaged TPM for the positions around the peak summits. To calculate the TPMs for this plot and to slightly smoothen it, reads were extended by 200 bp upstream and 350 bp downstream. The sum of this normalized signal was calculated for all positions in a ±2000 bp range around the peaks summits and divided by the total number of peaks, to produce the averaged normalized signal.

Genomic distribution: The percentage of peaks with their left most position overlapping with defined genomic locations was calculated and plotted. TSS and TES annotation was taken from Ensembl Genome database (Mus musculus, Rev. 83, mm10).

Heatmaps: To compare the signal enrichment in commonly reported peaks from SMARCAD1-FLAG and endogenous SMARCAD1 ChIP-Seq, 10,000 bp spanning regions around all of the shared 2380 peak regions were centered at the summit of the SMARCAD1-WT signal (TPM). To allow comparability of the lanes, all signals were normalized to TPM with the 98th percentile of the SMARCAD1-WT signal set as the maximum value. Darker colors indicate higher signal intensities.

Publicly available external ChIP-Seq data were obtained from GEO using the following accession numbers: GSM1555120 (KAP1), GSM1429923 (KAP1 input), GSM307622 (H4K20me3), GSM1033638 (H3K27me3), GSM594578 (H3K27ac), GSM1033636 (H3K4me3), GSM1555116 (H3.3), GSM459273 (SETDB1), GSM1215219 (G9a), GSM1375157 (SUV39H1), GSM1375158 (SUV39H2).

Genome wide data were visualized using the Integrative Genomics Viewer (IGV)[68]. Area proportional Venn Diagrams displaying overlap between peaks from different datasets were generated using eulerAPE[69].

For analysis of repetitive elements we adopted the method used by[34]. Aligment: Repeatmasker reported repetitive elements were downloaded from UCSC Table Browser (http://genome.ucsc.edu) for mm10 on 17 April 2017. The mm10 genome build was filtered to these regions to create repetitive element sequences. To compensate for repeat elements that might be too short for correct read alignment, all repeat sequences were expanded by one average read length (51 bp) at their

beginning and end, respectively. A bowtie index was created using bowtie-build and ChIP-seq reads were aligned against these repeat sequences using Bowtie (parameters: -k 1 and --best)[70]. Normalized read counts (TPM) were calculated for each repetitive element and used for further analysis. To identify enriched binding of repetitive elements, log2 fold-change over input was calculated for repeat classes and custom repeat categories. The boxplot depicts the log2 fold-changes over input for all regions belonging to each of the custom repeat categories and the repeat class with the highest (averaged) fold-change. The box- and jitterplot depicts the log2 fold-changes over the corresponding input for all repeat regions belonging to each of the custom repeat categories. In the boxplots center lines show the median, lower, and upper box lines correspond to 25th and 75th percentiles with the upper whiskers extending from the hinge to the largest value no further than 1.5 * IQR from the hinge (where IQR is the inter-quartile range, or distance between the first and third quartiles). The lower whiskers extend from the hinge to the smallest value at most 1.5 * IQR of the hinge. Data beyond the end of the whiskers are outlying points.

For the comparison of SMARCAD1 and KAP1 binding at different ERV families a published KAP1 ChIP-seq dataset[34] was processed as described above for our ChIP-seq dataset from XX PGK12.1 ESCs. As the datasets were mismatched on sex, repeats mapping to the sex chromosomes were removed. Data were filtered to retrotransposons, grouped by repeat class and sample and sequencing-depth normalized read counts were plotted. Correlations were calculated using Spearman's correlation.

**Statistical analysis**. Statistical significance was determined by a paired two-tailed Student's *t*-test. Sample sizes are provided in the figure legends.

**Reporting summary**. Further information on experimental design is available in the Nature Research Reporting Summary linked to this article.

## Data availability
Sequencing data are available in the ArrayExpress repository under accessions E-MTAB-7011 (KAP1 ChIP-seq in E14 ESCs), E-MTAB-7012 (SMARCAD1 and H3K9me3 in PGK12.1 ESCs), and E-MTAB-7014 (FLAG-SMARCAD1 ChIP-seq in E14 ESCs). The authors declare that all data supporting the findings of this study are available within the article and its supplementary information files or from the corresponding author upon reasonable request. The source data underlying Fig. 1a, b, g; Fig. 5a, b; Fig. 6c, g and Supplementary Figures 1f; 2h; 4f; 6c; 7c; 8b; 10b,c,f are provided in Supplementary Figure 11. A reporting summary for this article is available as a Supplementary Information file.

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

## Acknowledgements

This work was supported by the Deutsche Forschungsgemeinschaft, Transregional Collaborative Research Centre 81, to J.E.M., P.S, P.B. and T.S. We acknowledge Miguel Branco (Blizard Institute, UK) and Xu Han (Life Sciences Institute, Zhejiang University, China) for pKLO.1 plasmids. We are grateful to Katrin Treutwein, Tanja Kellermann, Tamina Rückert, Boris Klimovich, Florian Hub, Bianca Bamberger, and the Flow

Cytometry Core Facility Marburg for their excellent contributions to the acquisition of data. We thank Marek Bartkuhn, Alexander Brehm, Colin Dingwall, Lienhardt Schmitz, Guntram Suske and Marco Mernberger for helpful discussions.

## Author contributions

The study was conceived and designed by J.E.M., P.S., D.D and P.S., D.D., P.B., C.S., generated the data; A.N. and T.S. performed sequencing which was analysed by B.L. and F.F., J.M wrote the paper with input from all authors.

## Additional information

**Competing interests:** The authors declare no competing interests.

