## [Peer Review File · Nature Communications]

Reviewers' comments:

Reviewer #1 (Remarks to the Author):

SMARCAD1 ATPase activity is required to silence endogenous retroviruses
in embryonic stem cells
Parysatis Sachs1, et al

The authors provide evidence for a mechanism that maintains repression of ERV and their surrounding genes shown in Fig 7 that involves a repressive complex containing KAP, which appears to recruit SMARCAD1 and SETDB1 and HDAC. They find that this complex requires the ATPase activity to maintain repression and suggest possible mechanisms that might involve nucleosome remodeling. The study is well performed the results clear and the implications well supported. It is also clearly written. I have only a few suggestions:

In Figures 1 and 2 they validate their approach. However, one thing I found lacking was an investigation of whether the proteins involved exist in the cell as a true complex that could be characterized on a gradient analysis. This would be useful.

In Figure 4 they show that depletion of SMARCAD1 leads to expression of a subset of ERV's Do you have any information on the nature of this subset?

In figures 6B and C they kd SMARCAD1, but not the introduced ATPase mutant. Why were they both not reduced? Does the introduced copy contain a deletion of the binding site for the siRNA? This should be clarified.

It would be useful to know something about the specificity of SMARCAD1. Would knocking down other SNF2-like ATPases also induce the expression of the ERV and surrounding genes?

Does the ATPase mutation prevent a solution interaction with KAP?

Reviewer #2 (Remarks to the Author):

MS ID#: NCOMMS-18-24488

In this manuscript, the authors described that SWI/SNF-like chromatin remodeling factor SMARCAD1 plays a key role for endogenous retroviruses (ERVs) silencing mediated by the KAP1 pathway in mouse ES cells (mESCs). They showed that SMARCAD1 targets ERV subfamilies class I and II, particularly active intracisternal A-type particles (IAPs). Accumulation of the KAP1-SETDB1 complex and epigenetic silencing marks histone H3 lysine 9 trimethylation (H3K9me3) and H4K20me3 on the SMARCAD1-targeted ERVs are diminished in SMARCAD1 depleted mESCs, and IAPs are mildly derepressed. Furthermore, they showed that SMARCAD1's KAP1 binding is essential for its targeting to the ERVs, and SMARCAD1 and KAP1 occupancy at ERVs is co-dependent and requires the ATPase function of SMARCAD1.

In their original Mol Cell paper (Rowbotham et al 2011), they have already reported that SMARCAD1 is a key factor for the re-establishment of repressive chromatin, and global level of H3K9me2 and me3 are significantly diminished in SMARCAD1 knock down (KD) cells. They also showed that KAP1 is a major partner of SMARCAD1 as a complex, and pericentric H3K9me3 and KAP1 are significantly diminished in SMARCAD1 KD cells. Furthermore, they have shown that ATPase function of SMARCAD1 is essential for its role in H3K9me3 regulation. And, it is also already well known that the KAP1-

SETDB1 pathway plays a key role for ERV class I and II silencing in mESCs. Therefore, although the manuscript is well written and quality of their dataset is high, most of their findings are probably well predicted other than the KAP1 impaired ERV accumulation phenotype in SMARCAD1 KD mESCs complemented with ATPase defective mutant of SMARCAD1. Nevertheless, this work is still quite valuable for complete understanding of heterochromatin establishment and maintenance based on the view of chromatin remodeling/dynamics. Therefore, the reviewer is supportive to publish it in Nat Communication once the authors can respond to the following comments properly.

Major criticisms,

1. SMARCAD1 peaks vs KAP1 peaks on ERVs and how much difference between SMARCAD1 KO/KD vs KAP1 KO/KD for ERV derepression?

From Fig. 1F and S2G, it is obvious that most of SMARCA1 peaks are included in the KAP1 peaks. However, it is not clear that how much of SMARCAD1 peaks on ERVs are overlapped with KAP1 peaks on ERVs. The reviewer guesses that only part of KAP1 ERV-targets are also bound with SMARCAD1. The format shown in Fig. 1D in Maksakova et al. *Epigenetics & Chromatin* 2013, 6:15 is useful for comparison of SMARCAD1 peaks vs KAP1 peaks on ERVs. Furthermore, it is not clear in the current manuscript that how much difference between SMARCAD1 KO/KD vs KAP1 KO/KD for ERV derepression in mESCs. Only they showed that MMERVK10C behaved differently and MMERV10C is more derepressed in KAP1 KD. So, the expression heat map data of all kinds of ERVs for SMARCAD1 wt vs KD and KAP1 wt vs KD/KO are useful. The reviewer(/readers) want to know comprehensively how much SMARCAD1 contributes (or is critical) to KAP1-mediated ERV silencing in mESCs. The authors may argue that KAP1 KO or KD also diminishes SMARCAD1 function, but still such information is useful.

2. ChIP-qPCR and RT-qPCR analysis

The authors should perform statistics analysis for all their ChIP-qPCR and RT-qPCR data sets and indicate which differences are significant.

3. expression of nearby genes

Fig. 4C, 6D (left), S6B and S6C show the expression status of nearby genes of the derepressed IAPs by SMARCAD1 KD. But, it is not shown whether these up-regulated nearby genes are ERV fusion transcript (initiated from the derepressed IAP)? The authors should check it.

4. Fig. 6H

Fig. 6H shows that reduced KAP1 binding to its target ERVs is not rescued by ATPase mutant SMARCAD1. It is also nice to show how this phenotype can be rescued by the KAP1-binding defective mutant, CUE1 mt.

Minor comments,

5. Fig. 1A

XX and XY ES cells, the authors should indicate which one is PGK12.1 or E14 in the figure or figure legend.

6. Fig. 5B and Fig. 6D

CUE1 mutant which can't bind to KAP1, how about stability of protein is? In their recent JBC paper (Ding et al 2018), exogenous V5-tagged SMARCAD1 is also destabilized by KAP1 KD. To maintain WT level of expression, CUE1 mt need more transcription? Complex formation with KAP1 is essential for SMARCAD1 stability?

7. Fig. S1E

What means 2ug and 4ug? Loaded sample protein amounts? Should clarify in the legend.

Reviewer #3 (Remarks to the Author):

SMARCAD1 ATPase activity is required to silence endogenous retroviruses in embryonic stem cells

Sachs et al.

The authors identify SMARCAD1 to exert a critical role in the formation of the KAP1 repressive complex that is enriched at endogenous retroviruses (ERVs) in mouse embryonic stem cells. SMARCAD1 is required for silencing ERVs of the IAP class and co-regulated adjacent genes. SMARCAD1 interacts with KAP1 through its CUE domain, while its ATP-dependent chromatin remodeling function appears to be required to stabilise binding of SETDB1 and KAP1 and preserve histone methylation. I found this manuscript interesting and novel and very clearly written. The data were also beautifully presented. The main concern I have, however, is that there is very little effect of SMARCAD1 depletion on the transcription of ERVs and adjacent genes (see below).

Major comments

1. SMARCAD1 depletion appears to have little effect on the expression of ERVs and genes (Figure 4). KAP1 that was depleted in parallel has a similar only very modest effect on expression of ERVs, which contrasts with previous studies on KAP1 (references 13,17,24,44 for example). The authors should reassess the requirement of SMARCAD1 for ERV silencing using stable knockdown / knockout and a time-course to see if here the phenotype has been missed. If later time-points cannot be assessed due to lethality, this should be discussed. Also, since this is a key message of the paper, more ERV primers should be assessed or RNA-seq performed.

Minor comments

1. Figure 1C shows reduced proliferative capacity of the SMARCAD1 KD ESCs without alteration in cell cycle. However, it's not clear if SMARCAD1-depleted ESCs die or differentiate or both, please clarify. A cell viability assay would be helpful.
2. In the discussion, it's stated that "LINE elements did not emerge as significantly enriched SMARCAD1 targets in our ChIP experiments, but a detailed examination of discrete LINE sub-families has not been carried out." However, it should be mentioned that the young LINE1 subfamily L1Md_F was examined and found enriched with KAP1 and H3K9me3 but not SMARCAD1 (Figure 2).
3. Figure 3: It's curious that SMARCAD1 only affects binding of KAP1 to IAP elements where SMARCAD1 is most enriched and not of KAP1 to other ERVs. This should be clear in the text and in the model in Figure 7. In the model in Figure 7, it would be more accurate to change ERV 1/ II to "ERV II" or "IAP ERV" and shouldn't KAP1 be absent in B. and C.?
4. Figure 6 H: This is an important result but are differences significant here?
5. Figure 4D needs a legend for each gene examined.
6. Line 1000 for figure 1B - the phrase 'cell extracts' is missing a space.

Response to reviewers

We thank all reviewers for carefully considering our manuscript, for their insightful and positive comments and their constructive criticism. We have conducted additional experiments and prepared a revised version of our manuscript addressing all of the reviewers suggestions. We feel that the new data strengthen and complement our results and conclusions.

Reviewer #1 (reviewers comments in *italics*)

The authors provide evidence for a mechanism that maintains repression of ERV and their surrounding genes shown in Fig 7 that involves a repressive complex containing KAP, which appears to recruit SMARCAD1 and SETDB1 and HDAC. They find that this complex requires the ATPase activity to maintain repression and suggest possible mechanisms that might involve nucleosome remodeling. The study is well performed the results clear and the implications well supported. It is also clearly written. I have only a few suggestions:

1) *In Figures 1 and 2 they validate their approach. However, one thing I found lacking was an investigation of whether the proteins involved exist in the cell as a true complex that could be characterized on a gradient analysis. This would be useful.*

Response: We have several lines of evidence for the existence of a complex between SMARCAD1 and KAP1. Both proteins are highly abundant in the ESC nucleus and our biochemical data indicate that biochemically distinct complexes exist, a chromatin-bound complex and a soluble complex which is released from nuclei in the presence of low salt concentrations (Dignam et al., NAR 1983) and is stable in the absence of nucleic acids (this manuscript and Dong et al., 2018). We have now characterized the distribution profiles of SMARCAD1, KAP1 and SETDB1 in these nuclear extracts by gel filtration. This revealed that these proteins co-elute in fractions with a $MW_{app} > 600KDa$ (newly provided Supplementary Figure 2h and reference to this Figure in the text, last paragraph p9).

Moreover, endogenous SMARCAD1, KAP1 and SETDB1 can be co-immunoprecipitated from ESC extracts in the absence of nucleic acids (Figure 1g and Dong et al., JBC 2018; Thompson et al., PloS Genetics 2015). Further evidence supporting the existence of a SMARCAD1-KAP1 complex comes from the co-localization of these proteins in indirect immunofluorescence experiments, both at DAPI dense structures and throughout the nucleoplasm. Re-ChIP experiments presented in this manuscript (Figure 2c) demonstrate that SMARCAD1 and KAP1 co-occupy the same genomic targets. Collectively our data demonstrate that stable, soluble and chromatin bound complexes of these two proteins exist within the cell.

2) In Figure 4 they show that depletion of SMARCAD1 leads to expression of a subset of ERV's Do you have any information on the nature of this subset?

Response: We have expanded our expression analysis employing more ERV specific primers which are presented in Supplementary Figure 7a. It is evident that ERV classes including IAPs, MMERVK10C and ETn elements are affected by SMARCAD1 depletion, including a subgroup of IAPs belonging to the IAPEz int family.

3) In figures 6B and C they kd SMARCAD1, but not the introduced ATPase mutant. Why were they both not reduced? Dose the introduced copy contain a deletion of the binding site for the siRNA? This should be clarified.

Response: The introduced ATPase mutant was not reduced compared to the endogenous SMARCAD1 because the shRNA used to achieve knockdown targeted specifically the 3' UTR of SMARCAD1 which was absent from the ATPase mutant expression construct.

In the results we explain:

Epitope tagged wild-type (WT) and SMARCAD1 ATPase mutant (mt) expression vectors were stably introduced into ESCs carrying a doxycycline inducible shRNA targeting the 3'UTR of *Smarcad1* (Figure 6b). Endogenous SMARCAD1 protein was subsequently depleted and tagged SMARCAD1 proteins, WT and mutant, were expressed at comparable levels (Figures 6b and 6c).

4) It would be useful to know something about the specificity of SMARCAD1. Would knocking down other SNF2-like ATPases also induce the expression of the ERV and surrounding genes?

Response: This is an interesting question, also in the light that remodelers could act cooperatively or consecutively to implement required chromatin changes. SNF2 type factors like CHD5 and ATRX have previously been implicated in ERV regulation in mESCs. To our knowledge, none of these ATPases has been reported to interact directly with KAP1, as we have observed for SMARCAD1. We show that SMARCAD1 co-operates with KAP1, binds to ERVs of class I and II and effects their expression, especially of IAPs. For ATRX, linked to heterochromatin formation at ERVs, the impact on ERV expression is not clear; some papers report no change of IAP expression in its absence (Hoelper et al. 2017, *Nat commun.* 8:1193; Sadic et al., 2015, *EMBO J.* 6(7): 836–850) while others report inappropriate upregulation of IAPs (He et al., 2015, *Cell Stem Cells* Vol17;3:273-286; Robbez-Masson et al., 2018, *Genome Res* 28(6):836-845). Studies of CHD5 revealed specificity in SNF2 like ATPases mediated transcriptional control of retrotransposons (Hayashi et al. 2016, *Journal of cellular biochemistry*, 117:780-792). Disruption of CHD5 impacts the expression of class III ERV elements (MERV1), but not other ERV classes (e.g. IAPs). Moreover, *Cml2* and *Bglap3* genes, which our experiments highlight as considerably de-repressed upon SMARCAD1 loss, were not affected by CHD5 loss.

In the introduction we review the role of other SNF2 like factors at ERVs as follows:

In pluripotent stem cells SNF2 helicase family members such as CHD5 (chromodomainhelicase DNA binding protein 5) and ATRX (athalassaemia/mental retardation syndrome X-linked) have been implicated in the control of class III MERV1 and class II IAP elements respectively.

5) Does the ATPase mutation prevent a solution interaction with KAP1?

Response: No, the ATPase mutation in SMARCD1 does not prevent a solution interaction with KAP1 based on (a) GST pulldowns with purified components (newly provided Supplementary Fig.9f), (b) biochemical fractionation of the ATPase mutant (newly provided Supplementary Fig.9b) and (c) co-immunoprecipitations in cells (Fig.6g).

(a) We have performed interaction studies with recombinant, purified proteins *in vitro*, utilizing bacterially expressed GST-KAP1 protein and SMARCD1 proteins produced in TNT(T7) system. In GST-pull down assays shown in Supplementary Fig. 9f the interaction of SMARCD1 ATPase mutant with GST-KAP1 protein is comparable to WT SMARCD1 protein (compare lanes 4 and 6). This is consistent with our earlier work in which we showed that the KAP1-SMARCD1 interaction is mediated by the CUE1 domain alone.

(b) As the editor asked for further characterization of the ATPase mutant we show that SMARCD1 ATPase mutant behaves as the wildtype protein in cellular fractionation experiments. We performed Western blot analysis of ESCs expressing tagged SMARCD1, either WT or ATPase mutant, fractionated into chromatin enriched and soluble fractions and found that - along with KAP1 - both WT and mutant SMARCD1 are predominantly in the soluble fraction (Supplementary Fig. 9b).

(c) Figure 6g addresses the same question in the form of an immunoprecipitation experiment of either epitope tagged WT or mutant SMARCD1 protein and subsequent evaluation of the quantity of co-immunoprecipitated KAP1. Shown is a FLAG immunoprecipitation in the same cell lines used for ChIP-seq and ChIP-qPCR analysis of the ATPase mutant upon depletion of endogenous SMARCD1 protein (WT FLAG SMARCD1 lanes 1-3; FLAG-tagged ATPase mutant SMARCD1 lanes 4-6). Western blot analyses indicated that equivalent amounts of wild type and mutant FLAG-SMARCD1 were recovered in immunoprecipitations (Fig. 6g, FLAG Western, compare lanes 3 with 6). The association of KAP1 with the mutant SMARCD1 protein was not affected compared to the wild type protein (Fig. 6g, KAP1 Western, compare lane 3 with 6).

These co-immunoprecipitations were performed in the presence of benzonase and ethidiumbromide to eliminate interactions mediated by DNA. Thus, an intact ATPase domain is not important for the efficient interaction of SMARCD1 with KAP1 in cells.

Reviewer #2 (reviewers comments in *italics*)

In this manuscript, the authors described that SWI/SNF-like chromatin remodeling factor SMARCAD1 plays a key role for endogenous retroviruses (ERVs) silencing mediated by the KAP1 pathway in mouse ES cells (mESCs). They showed that SMARCAD1 targets ERV subfamilies class I and II, particularly active intracisternal A-type particles (IAPs). Accumulation of the KAP1-SETDB1 complex and epigenetic silencing marks histone H3 lysine 9 trimethylation (H3K9me3) and H4K20me3 on the SMARCAD1-targeted ERVs are diminished in SMARCAD1 depleted mESCs, and IAPs are mildly derepressed. Furthermore, they showed that SMARCAD1's KAP1 binding is essential for its targeting to the ERVs, and SMARCAD1 and KAP1 occupancy at ERVs is co-dependent and requires the ATPase function of SMARCAD1.

In their original Mol Cell paper (Rowbotham et al 2011), they have already reported that SMARCAD1 is a key factor for the re-establishment of repressive chromatin, and global level of H3K9me2 and me3 are significantly diminished in SMARCAD1 knock down (KD) cells. They also showed that KAP1 is a major partner of SMARCAD1 as a complex, and pericentric H3K9me3 and KAP1 are significantly diminished in SMARCAD1 KD cells. Furthermore, they have shown that ATPase function of SMARCAD1 is essential for its role in H3K9me3 regulation. And, it is also already well known that the KAP1-SETDB1 pathway plays a key role for ERV class I and II silencing in mESCs. Therefore, although the manuscript is well written and quality of their dataset is high, most of their findings are probably well predicted other than the KAP1 impaired ERV accumulation phenotype in SMARCAD1 KD mESCs complemented with ATPase defective mutant of SMARCAD1.

Nevertheless, this work is still quite valuable for complete understanding of heterochromatin establishment and maintenance based on the view of chromatin remodeling/dynamics. Therefore, the reviewer is supportive to publish it in Nat Communication once the authors can respond to the following comments properly.

Major criticisms:

1. SMARCAD1 peaks vs KAP1 peaks on ERVs and how much difference between SMARCAD1 KO/KD vs KAP1 KO/KD for ERV derepression?

From Fig. 1F and S2G, it is obvious that most of SMARCA1 peaks are included in the KAP1 peaks. However, it is not clear that how much of SMARCAD1 peaks on ERVs are overlapped with KAP1 peaks on ERVs. The reviewer guesses that only part of KAP1 ERV-targets are also bound with SMARCAD1. The format shown in Fig. 1D in Maksakova et al. Epigenetics & Chromatin 2013, 6:15 is useful for comparison of SMARCAD1 peaks vs KAP1 peaks on ERVs. Furthermore, it is not clear in the current manuscript that how much difference between SMARCAD1 KO/KD vs KAP1 KO/KD for ERV derepression in mESCs. Only they showed that MMERVK10C behaved differently and MMERV10C is more derepressed in KAP1 KD. So, the expression heat map data of all kinds of ERVs for SMARCAD1 wt vs KD and KAP1 wt vs KD/KO are useful. The reviewer(/readers) want to know comprehensively how much SMARCAD1 contributes (or is critical) to KAP1-mediated ERV silencing in mESCs. The authors may argue that KAP1 KO or KD also diminishes SMARCAD1 function, but still such information is useful.

Response “Peaks”: We thank the reviewer for this suggestion and performed the requested analysis. We compared median signals of SMARCAD1 /KAP1 peaks over

input across retrotransposons. In the newly provided Figure 2b, class I, II and III ERV families are presented in different colours, revealing that the biggest correlation between SMARCAD1 and KAP1 binding is at IAPs (black triangles), followed by other class II elements (e.g. RLTR27). Interestingly, few differences are apparent between the individual subtypes of retrotransposons bound by KAP1 and SMARCAD1. For instance, both proteins are similarly enriched at IAPEz int but class III MaLR is modestly enriched for SMARCAD1 and not for KAP1. The new Figure 2b replaces Figure 2b of the original manuscript (a genome browser screenshot which is now incorporated in the Supplementary Figure 3 as panel b). A binding comparison of SMARCAD1 and KAP1 over all members of individual retrotransposon families (same families as in Fig2a) is additionally provided as Supplementary Fig. 3c.

Response “Expression”: To further address the relative contribution of SMARCAD1 and KAP1 KD on ERV expression we have examined additional ERV primers. These new RT-qPCR results have been incorporated into Figure 4.

Additionally, we now provide expression heatmap data for SMARCAD1 control vs KD and KAP1 control vs KD (new Supplementary Figure 6d). Data was collected at two different timepoints after induction of KD, 3 days and 5 days.

Immunoblot analysis shown next to the heatmap (Supplementary Figure 6d) demonstrates that KAP1 depletion for 3 and 5 days is accompanied by a progressive fall in SMARCAD1 protein levels. Hence, KAP1 KD impacts not only on KAP1 function but has consequences for SMARCAD1 function (as indicated by the reviewer). It is therefore not possible to unambiguously determine the extent to which the observed pronounced increase in ERV transcription at 5d KAP1 KD cells is attributable to KAP1 or to combined SMARCAD1/KAP1 function.

Is SMARCAD1 itself critical for ERV silencing? We conclude that it is, as the effect of individual depletion of SMARCAD1 on specific ERVs and nearby genes is a 2-8 fold transcriptional up-regulation compared to Ctrl KD (Figure 4 and Supplementary Figures 6 b,c). This level of up-regulation does not substantially increase between day 3 and day 5. Importantly, de-repression induced by SMARCAD1 KD can be reversed upon re-introducing WT SMARCAD1 (Figure 4c). This demonstrates that SMARCAD1 is required for repression of ERVs. Increased expression changes observed in cells where both KAP1 and SMARCAD1 are effectively depleted (KAP1 KD day 5) reveal that for maximum levels of ERV reactivation, KAP1 KD is needed, further strengthening our conclusion that there is a functional interdependence between SMARCAD1 and KAP1 in ERV regulation.

2. ChIP-qPCR and RT-qPCR analysis

The authors should perform statistics analysis for all their ChIP-qPCR and RT-qPCR data sets and indicate which differences are significant.

Response: We have now performed statistical analysis for ChIP-qPCR data provided in Figs.3a, 3b, 5b, 6f, 6h and Supplementary Figures 5c, 8b, 9d, 9e, 9h. Reviewer 2 similarly raised the question whether the reduced KAP1 binding at ERVs observed when the ATPase function of SMARCAD1 is compromised (Fig. 6h) is significant. Our newly provided statistical analysis reveals this to be the case.

Fig.4 shows RT-qPCR data collected following SMARCAD1 depletion for 3 days. Originally we presented biological duplicates, we have now extended these data sets by incorporating more ERV elements in our analysis and by performing more independent biological replicates: n=5 for one SMARCAD1 shRNA and n=4 for the second SMARCAD1 shRNA. This allowed us to perform statistical analysis. We found the majority of expression changes to be significant, in line with our conclusions.

3. expression of nearby genes

Fig. 4C, 6D (left), S6B and S6C show the expression status of nearby genes of the derepressed IAPs by SMARCAD1 KD. But, it is not shown whether these up-regulated nearby genes are ERV fusion transcript (initiated from the derepressed IAP)? The authors should check it.

Response: We have now verified the presence of chimaeric transcripts between promoter-proximal ERVs and downstream genes in SMARCAD1 KD cells by performing additional qPCR analysis. Previously characterized primers for *Cyp2b23* (Karimi et al., 2011 *Cell Stem Cell* 8(6): 676-87) and newly designed primers for *Cml2* that recognize an ERV element 5' to the annotated genes and an exon within the genes revealed up-regulation in SMARCAD1 KD compared to Ctrl KD cells. These results are included in Supplementary Figure 6b (XX ESCs PGK12.1) and 6c (XY ESCs E14 2i growth conditions) and demonstrate that these SMARCAD1 regulated genes are ERV fusion transcripts (see also new Supplementary Fig. 7a for a schematic of the primer location and new Supplementary Figure 7b for the size analysis of the obtained PCR products).

4. Fig. 6H

Fig. 6H shows that reduced KAP1 binding to its target ERVs is not rescued by ATPase mutant SMARCAD1. It is also nice to show how this phenotype can be rescued by the KAP1-binding defective mutant, CUE1 mt.

Response: Following up on this suggestion we now provide new data that the reduced KAP1 binding to its target ERVs observed upon SMARCAD1 depletion is indeed rescued by the KAP1-binding defective SMARCAD1 (CUE1 mt). Specifically, this observation has been incorporated into Figure 6h, and the corresponding Supplemental Figure 9h.

Minor comments,

5. Fig. 1A XX and XY ES cells, the authors should indicate which one is PGK12.1 or E14 in the figure or figure legend.

Done.

6. Fig. 5B and Fig. 6D

CUE1 mutant which can't bind to KAP1, how about stability of protein is? In their recent JBC paper (Ding et al 2018), exogenous V5-tagged SMARCAD1 is also destabilized by KAP1 KD. To maintain WT level of expression, CUE1 mt need more transcription? Complex formation with KAP1 is essential for SMARCAD1 stability?

Response: In the current manuscript we have a situation in which KAP1 protein is present but the interaction of SMARCAD1 with KAP1 is disrupted. The reviewer asks whether SMARCAD1 protein stability or mRNA levels are affected under these conditions. We find CUE1 mutant SMARCAD1 protein levels are comparable to WT SMARCAD1 protein levels in our experiments (e.g. Supplementary Figure 9c). We have carried out additional experiments to examine the RNA expression levels of the CUE1 mutant compared to the WT (see below, primers detect only tagged but not endogenous SMARCAD1) and they are very similar. This argues against the model that increased transcription of the CUE1 mutant is needed to maintain WT levels of protein.

Expression analysis detecting FLAG SMARCAD1

We consistently observe that depletion of KAP1 results in concomitant and progressive reduction of SMARCAD1 protein levels and also SMARCAD1 mRNA levels (Dong et al., 2018 and the current manuscript). The underlying mechanisms are not clear, except that this reduction is not simply an indirect consequence of cell differentiation and its associated decline in the pluripotency TF network which regulates SMARCAD1 expression. Evidence: Tagged SMARCAD1 expressed from a promoter no longer under the control of the pluripotency network is similarly unstable upon KAP1 loss.

Collectively, our data to date show that SMARCAD1 protein stability is affected when KAP1 protein is absent but not when KAP1 protein levels are normal but the interaction with SMARCAD1 is disrupted. We speculate that normal levels of KAP1 protein are required to maintain normal SMARCAD1 steady state levels but that complex formation is not crucial.

7. Fig. S1E *What means 2ug and 4ug? Loaded sample protein amounts? Should clarify in the legend.*

This refers to total protein amount loaded and is now clarified in the figure and the legend.

Reviewer #3 (reviewers comments in *italics*)

The authors identify SMARCAD1 to exert a critical role in the formation of the KAP1 repressive complex that is enriched at endogenous retroviruses (ERVs) in mouse embryonic stem cells. SMARCAD1 is required for silencing ERVs of the IAP class and co-regulated adjacent genes. SMARCAD1 interacts with KAP1 through its CUE domain, while its ATP-dependent chromatin remodeling function appears to be required to stabilise binding of SETDB1 and KAP1 and preserve histone methylation.

I found this manuscript interesting and novel and very clearly written. The data were also beautifully presented. The main concern I have, however, is that there is very little effect of SMARCAD1 depletion on the transcription of ERVs and adjacent genes (see below).

We thank the reviewer for the supportive comments and helpful suggestions.

Major comments

1. SMARCAD1 depletion appears to have little effect on the expression of ERVs and genes (Figure 4). KAP1 that was depleted in parallel has a similar only very modest effect on expression of ERVs, which contrasts with previous studies on KAP1 (references 13,17,24,44 for example). The authors should reassess the requirement of SMARCAD1 for ERV silencing using stable knockdown / knockout and a time-course to see if here the phenotype has been missed. If later time-points cannot be assessed due to lethality, this should be discussed. Also, since this is a key message of the paper, more ERV primers should be assessed or RNA-seq performed.

Response divided into the following sections

KAP1 depletion and its effect on ERV expression as seen in Figure 4

SMARCAD1 depletion and its effect on ERV expression as seen in Fig. 4

"More ERV primers should be assessed or RNA-seq performed"

Stable removal of SMARCAD1

Timecourse removal of SMARCAD1

Response KAP1 depletion and its effect on ERV expression as seen in Figure 4:

The reviewer is right that the effect on ERV expression upon KAP1 KD shown in Fig.4 is modest. However, this does not actually contrast with previous studies on KAP1 but is a consequence of the early time point that we have chosen to analyse. The Figure we provide below illustrates that the degree of transcriptional upregulation upon KAP1 KD is time dependent. Fig.4 shows data from day 3 after transfection, yet in general other reports present data collected later. We note that at these later timespoints usually presented in the literature, we do obtain up-regulation in a range comparable to previous studies.

KAP1 KD: Degree of transcriptional upregulation is time dependent

The rationale to present data collected at day 3 in Fig.4 is:

- We set out to understand the individual contributions of SMARCD1 and KAP1 to the repression of ERVs. However, as reported in our previous paper (Dong et al., 2018), KD of KAP1 results in progressive reduction of SMARCD1 protein levels. Day 3 was chosen to minimize the effect on SMARCD1 protein levels.
- We wish to understand the initial steps of how SMARCD1 and KAP1 depletion effects ERV transcription.
- At later timepoints, KAP1 KD cells appear quite differentiated.

In the revision we now include expression heat map data for SMARCD1 Control vs KD and KAP1 control vs KD encompassing data points for 3 days and 5 days after transfection (Supplementary Figure 6d). This will allow the reader to evaluate that in our hands KAP1 depletion has a similar effect on expression of ERVs as in previous studies.

Response SMARCD1 depletion and its effect on ERV expression as seen in Fig. 4:

We have now increased the number of biological replicates in Figure 4 to n=4 (shRNA1) and n=5 (shRNA2) and performed a statistical analysis which revealed that most changes are significant. Furthermore, we have increased the number of analysed sites in Fig.4 as well as in the complementary expression analysis performed in a different ESC line (Supplemental Figure 6b) and under different growth conditions (Supplemental Figure 6c). The effect of SMARCD1 depletion on specific ERVs and nearby genes is a 2-8 fold transcriptional up-regulation compared to Ctrl KD.

Collectively, our expression analysis supports a role of SMARCD1 in the regulation of ERV expression. This is further strengthened by the observation that transcriptional increase observed upon SMARCD1 loss can be rescued by re-expression of SMARCD1 (Figure 4c).

Response "More ERV primers should be assessed or RNA-seq performed":

We have expanded our expression analysis by systematically assessing more primers, listed below. Newly produced datasets have been added to Fig. 4 and Supplementary Figures 6 and 7.

- ERV consensus primers recognizing class I elements (VL30)
- More consensus primers for class II elements (IAP 5'UTR; IAPez-gag; MMERVK10C envelope)
- ERV consensus primer recognizing another class III element (MTA)
- More primers specific to individual SMARCAD1 bound ERVs, including IAPs (Rgs20), MMERVK10C elements (Chromosome 5; ZFP992, Rgs20) and an ETn element upstream of the *Cml2* gene
- We analysed the transcription of four additional nearby genes.
- Moreover, we strengthened our conclusions by confirming the existence of *Smarcad1* knockdown induced chimaeric transcripts between promoter proximal ERVs and their downstream genes (*Cml2*; *Cybp2b23*).
- Additionally, as mentioned above, we have performed 2-3 more biological replicates for the samples presented in Fig.4 to allow statistical analysis and we now indicate which differences are significant (most of them are).

Taken together, mainly members of the ERV II class of transposable elements were up-regulated upon SMARCAD1 knockdown, and also VL30, a member of class I, in agreement with the ChIP-seq data.

We appreciate the concern that a phenotype may have been missed and performed additional experiments to investigate this further.

Response to Stable removal of SMARCAD1:

To understand the initial process of how SMARCAD1 depletion is involved in expression changes of ERVs we have performed RNA expression analysis upon transient knockdown of SMARCAD1. Stable knockdown cells, which lack a particular protein for an extended period, often exhibit adaptive, compensatory changes. Moreover, a direct comparison of results obtained upon stable SMARCAD1 KD with stable KAP1 KD is not possible as KAP1 knockdown results in arrest of cell proliferation, precluding the generation of stable KAP1 KD ESC lines.

With these caveats in mind we have followed the reviewers suggestion and assayed ERV transcription in stable SMARCAD1 KD cells. Two independent clonal stable SMARCAD1 knockdown ESC lines showed no up-regulation of class I and II ERVs (see below), while the same primer sets showed a moderate but reproducible up-regulation upon transient SMARCAD1 KD. Given the likelihood of adaptation in cells that have experienced loss of SMARCAD1 over a longer time-period we do not feel confident about drawing any strong conclusions and prefer not to include these data in the manuscript.

Response to Timecourse removal of SMARCAD1:

We want to thank the reviewer for this suggestion. One, we addressed whether we are missing a spike of up-regulation very early on and found this not to be the case. We compared ERV expression following 2d or 4d of SMARCAD1 KD and found that effects on ERV expression are not yet apparent 2 days after transfection.

Analysis of ERV expression two days (grey) or 4 days (blue) after SMARCAD1 KD

We further examined the effect of SMARCAD1 depletion on ERV and gene transcription at later timepoints, three and five days after induction of KD. In parallel we performed the same analysis upon KAP1 depletion in order to dissect the relative contribution of these protein partners to ERV regulation. Expression heatmap data for SMARCAD1 control vs KD and KAP1 control vs KD has been provided as new Supplementary Figure 6d. We have extensively discussed this data in our response to Reviewer 2 Point 1, which we would like to repeat here.

The level of up-regulation caused by SMARCAD1 does not substantially change between day 3 and day 5. This contrasts with the effect caused by KAP1 KD, where we observe a progressive increase for the majority of loci analysed. Immunoblot analysis shown next to the heatmap demonstrates that KAP1 depletion for 3 and 5 days is accompanied by a progressive fall in SMARCAD1 protein levels. Hence, KAP1 KD impacts not only on KAP1 function but also diminishes SMARCAD1 function. It is therefore not possible to unambiguously determine the extent to which the observed pronounced increase in ERV transcription at 5d KAP1 KD cells is attributable to KAP1 or to combined SMARCAD1/KAP1 function.

De-repression induced by SMARCAD1 KD can be reversed upon re-introducing WT SMARCAD1, demonstrating that SMARCAD1 is required for repression of ERVs. Increased expression changes observed in cells where both KAP1 and SMARCAD1 are effectively depleted (KAP1 KD day 5) reveal that for maximum levels of ERV reactivation, KAP1 KD is needed, further strengthening our conclusion that there is a functional interdependence between SMARCAD1 and KAP1 in ERV regulation.

Minor comments

1. Figure 1C shows reduced proliferative capacity of the SMARCAD1 KD ESCs without alteration in cell cycle. However, it's not clear if SMARCAD1-depleted ESCs die or differentiate or both, please clarify. A cell viability assay would be helpful.

Response: Reduced proliferation in SMARCAD1 KD ESCs was not associated with induction of massive cell death compared to control KD, as determined by live cell imaging over 3 days in two different ESC lines (PGK12.1 and E14 ESCs). Representative time lapse videos acquired with IncuCyte are now included as newly provided Supplementary Movies 1 and 2, illustrating that SMARCAD1 KD ESCs do not display overt apoptosis but rather flattened morphology. As a measure of how much SMARCAD1 KD ESCs deviate from the typical ESC colony shape, eccentricity was quantified in real time. These new datasets have been added as Supplementary Figure 1e.

2. In the discussion, its stated that “LINE elements did not emerge as significantly enriched SMARCAD1 targets in our ChIP experiments, but a detailed examination of discrete LINE sub-families has not been carried out.” However, it should be mentioned that the young LINE1 subfamily L1Md_F was examined and found enriched with KAP1 and H3K9me3 but not SMARCAD1 (Figure 2).

Response: The reviewer is obviously right in that we have investigated SMARCAD1 binding at one young LINE1 subfamily L1Md_F. But as the focus of the paper is on ERVs we did not explore this further and did not test systematically different subfamilies of LINE elements under different conditions. We would therefore prefer not to overemphasize this observation and removed the sentence in the discussion outlined above by the reviewer.

3. Figure 3: It's curious that SMARCAD1 only affects binding of KAP1 to IAP elements where SMARCAD1 is most enriched and not of KAP1 to other ERVs. This should be clear in the text and in the model in Figure 7. In the model in Figure 7, it would be more accurate to change ERV I/ II to “ERV II” or “IAP ERV” and shouldn't KAP1 be absent in B. and C.?

Response: While the effect on KAP1 binding upon SMARCAD1 depletion is indeed the most significant at IAP elements, it is not limited to IAP elements. The reviewer refers to Figure 3; it is correct that in PGK12.1 ESCs stably depleted for SMARCAD1, KAP1 binding is mainly affected at IAPs. In E14 ESCs transiently depleted for SMARCAD1 this is similarly the case, however, other ERV families are affected as well. This is apparent from our KAP1 ChIP-seq analysis upon SMARCAD1 KD (Supplemental Figure 5a). We have confirmed a significant

reduction of KAP1 at MMERVK10C by ChIP qPCR in Supplementary Fig.3c (upon SMARCAD1 depletion) and in Fig.6h (in the ATPase mutant).

We have indicated this in the original text, result section: “ChIP-qPCR analysis following a two-day depletion of SMARCAD1 moreover showed a subtle, yet reproducible reduction of KAP1 at other ERV families (Supplementary Figure 5c). “

Model in Fig.7.: We depict KAP1 in panels b, c to illustrate that KAP1 protein levels are not affected upon SMARCAD1 KD or in the ATPase mutant. To emphasize this further we have modified the legend of panel b which now reads “ In the absence of SMARCAD1, KAP1 *protein levels are not affected but its* binding to ERVs and the recruitment of SETDB1 are compromised”. We have also increased the physical separation of KAP1 from the ERV in the figure to make it clearer that KAP1 binding is affected.

With respect to the labelling of the cartoon with either ERV I/II or ERV II we want to draw the reviewers attention to data in support for a role of SMARCAD1 at ERVs of class I: WT SMARCAD1 is enriched at ERV elements belonging to class I (Fig.2a), SMARCAD1 and KAP1 binding is detected at class I VL30 element in re-ChIP experiments (Fig.2c) and upon depletion of SMARCAD1/KAP1 the expression of VL30 is increased (newly provided panels in Fig.4b and Supplementary Figures 6b,c). We therefore believe our model depicting SMARCAD1 functioning at ERVs of both class II and I accurately encapsulates our observations.

4. Figure 6 H: This is an important result but are differences significant here?

Response: Yes, these differences are statistical significant as determined by a Students *t* Test. A new Figure 6h - showing the mean of 3 biological replicates (rather than a representative example as before) and indicating the statistical significance - replaces Figure 6h of the original manuscript.

5. Figure 4D needs a legend for each gene examined.

We have improved that, thanks.

6. Line 1000 for figure 1B - the phrase 'cell extracts' is missing a space.

Amended.

REVIEWERS' COMMENTS:

Reviewer #1 (Remarks to the Author):

The authors have responded to my comments and the comments of other reviewers with new data and explanations. I feel that my objections are addressed, with the exception of Supp Fig 2h showing that SMARCAD1, KAP1 and SETDB1 are truly part of what might be called a biochemical complex. In fact, Figs2h shows that very little of SMARCAD1 comigrates on gradients with KAP1 and SETDB1. I feel the authors need to adjust their terminology to say that these proteins associate. Whether or not there is truly a complex, for example like the ribosome with dedicated subunits that can not be dissociated except by denaturation is unclear, indeed it is probably not a complex in the classic biochemical sense. This is a minor objection to what is an excellent paper.

Reviewer #2 (Remarks to the Author):

The reviewer satisfied with the authors' responses.
No further comments.

Reviewer #3 (Remarks to the Author):

The authors have thoroughly addressed all of my questions in their revised manuscript. I think this elegant work is suitable for publication in Nature Communications provided that the authors tone down their statements on the requirement of SMARCAD1 for ERV repression since its depletion exerts only a modest effect on ERV transcription.

I think that the KAP1 KD time course provided in the rebuttal letter would be informative to include in supplementary data. The authors could use this to explain that day 3 was focused on to avoid any cell differentiation / SMARCAD1 levels decreasing in the KAP1 KD.

However, the readers want to know about the requirement for SMARCAD1 in ERV repression and its own depletion causes little effect (Figure 4, Supplementary Figure 6d), although the complementation experiments (Figure 4c) and changes in histone methylation (Figure 3) are convincing. The further ERV consensus primers the authors have included confirm that there is not much effect on ERV expression following SMARCAD1 depletion (Figure 4). However, SMARCAD1 appears necessary to regulate genes nearby ERVs (Figure 4b, bottom panel), which should be emphasized. In the rebuttal letter, the authors provide a graph showing the role of SMARCAD1 in ERV repression / repression of genes near ERVs (day 2 and day 4, grey and blue bars) that is more convincing than the actual data presented in Figure 4. I suggest this data be substituted into Figure 4 to support the main conclusions of the manuscript.

Figure 4b: The label 10C Rgs20 should be labelled as "ERVK10C Rgs20".

NCCOMMS-18-24488 Sachs et al.
SMARCAD1 ATPase activity is required to silence endogenous retroviruses in ESCs

Dear Reviewers,

We have received your latest reviews for our manuscript NCCOMMS-18-24488 and want to thank you for your time and attention and for supporting the publication, in principle, of our work in *Nature Communications*.

Sincerely,

Jacqueline E. Mermoud

Response to reviewers

Reviewer #1 (reviewers comments in *italics*)

The authors have responded to my comments and the comments of other reviewers with new data and explanations. I feel that my objections are addressed, with the exception of Supp Fig 2h showing that SMARCAD1, KAP1 and SETDB1 are truly part of what might be called a biochemical complex. In fact, Figs2h shows that very little of SMARCAD1 comigrates on gradients with KAP1 and SETDB1. I feel the authors need to adjust their terminology to say that these proteins associate. Whether or not there is truly a complex, for example like the ribosome with dedicated subunits that can not be dissociated except by denaturation is unclear, indeed it is probably not a complex in the classic biochemical sense. This is a minor objection to what is an excellent paper.

We absolutely agree with the reviewer that we have not defined a SMARCAD1 complex in terms of dedicated and signature subunits as has been described for example for the BAF chromatin remodeller complex. Therefore we have been careful not to use the word complex in the legend to Sup Fig 2h or in the text (page 8) but have described the biochemical behaviour of the key proteins as 'co-elution'. We restrict our use of the word complex to describe the set of proteins which can be isolated by chromatin immunoprecipitation or immunoprecipitation of soluble proteins from nuclear extracts.

To help clarify this further we have introduced two further changes in the text:

(1) The introduction to the KAP1-SMARCAD1 interaction (page 5) was changed from "Our proteomic analysis identified KAP1 as a stoichiometric component of SMARCAD1 complexes in mouse ESCs (mESCs)."

to

"Our proteomics analysis revealed KAP1 to be robustly associated with SMARCAD1 in mouse ES cells."

(2) In the discussion the sentence “Our findings present a framework in which the function of KAP1-SMARCAD1 complexes can be dissected elsewhere in the genome.”

to

“Our findings present a framework in which KAP1-SMARCAD1 function can be dissected elsewhere in the genome.”

Reviewer #2 (reviewers comments in italics)

The reviewer satisfied with the authors' responses.

No further comments.

Reviewer #3 (reviewers comments in italics)

The authors have thoroughly addressed all of my questions in their revised manuscript. I think this elegant work is suitable for publication in Nature Communications provided that the authors tone down their statements on the requirement of SMARCAD1 for ERV repression since its depletion exerts only a modest effect on ERV transcription.

I think that the KAP1 KD time course provided in the rebuttal letter would be informative to include in supplementary data. The authors could use this to explain that day 3 was focused on to avoid any cell differentiation / SMARCAD1 levels decreasing in the KAP1 KD.

However, the readers want to know about the requirement for SMARCAD1 in ERV repression and its own depletion causes little effect (Figure 4, Supplementary Figure 6d), although the complementation experiments (Figure 4c) and changes in histone methylation (Figure 3) are convincing. The further ERV consensus primers the authors have included confirm that there is not much effect on ERV expression following SMARCAD1 depletion (Figure 4). However, SMARCAD1 appears necessary to regulate genes nearby ERVs (Figure 4b, bottom panel), which should be emphasized. In the rebuttal letter, the authors provide a graph showing the role of SMARCAD1 in ERV repression / repression of genes near ERVs (day 2 and day 4, grey and blue bars) that is more convincing than the actual data presented in Figure 4. I suggest this data be substituted into Figure 4 to support the main conclusions of the manuscript.

We have introduced changes throughout the manuscript to tone down the effect of SMARCAD1 on ERV transcription and emphasize the effect on nearby genes.

For instance, in the abstract, where we first introduce the results of this paper, we have now adjusted the wording such that we identify SMARCAD1 as a key factor in the control of ERVs in embryonic stem cells (rather than in their transcriptional repression). In the introductions last paragraph we have removed the sentence that SMARCAD1 depletion leads to activation of ERVs so that the emphasis of this paragraph has shifted to SMARCAD1 being a component of the machinery that silences ERVs, involved in setting up the stable association of KAP1. Similarly, in the first paragraph of the discussion and in the final Figure (the Model) the emphasis is on SMARCAD1 facilitating stable KAP1-SMARCAD1 binding at ERVs.

In the discussion we rephrased “Our expression studies showed that SMARCAD1 depletion recapitulates the misregulation of ERVs and nearby cellular genes observed upon KAP1 loss” to “Our expression studies showed that SMARCAD1 depletion not only has an effect on ERVs, but also results in misregulation of nearby cellular genes.”

Following the reviewers suggestions we now incorporate two figures supplied originally with the rebuttal letter in the manuscript.

The KAP1 KD timecourse is shown in the newly provided Supplementary Figure 6b and, as suggested, used to explain the rationale why 3 day (presented in Figure 4) was chosen for further analysis. The reviewer found the day 2 and day 4 comparative analysis of SMARCAD1 knockdown effects convincing, and we now provide this data as new Supplementary Figure 6a. These experiments are essential to understand why day 3 was chosen for further analysis (as 3 day is the earliest timepoint where expression changes are observed). We therefore believe that while the day2/day 4 analysis is ideal to complement Figure 4, it should not replace it. Not least because Figure 4 presents a more comprehensive and complete analysis that incorporates important controls, such as differentiation and statistical significance, and a direct comparison between SMARCAD1 and KAP1 transcriptional effects over a larger number of loci.

Figure 4b: The label 10C Rgs20 should be labelled as “ERVK10C Rgs20”.

Done.